# Concerted action of neuroepithelial basal shrinkage and active epithelial migration ensures efficient optic cup morphogenesis

Jaydeep Sidhaye[1,2], Caren Norden[1]*

[1]Max Planck Institute of Molecular Cell Biology and Genetics, Dresden, Germany;
[2]Dresden International Graduate School for Biomedicine and Bioengineering, Technische Universität Dresden, Dresden, Germany

**Abstract** Organ formation is a multi-scale event that involves changes at the intracellular, cellular and tissue level. Organogenesis often starts with the formation of characteristically shaped organ precursors. However, the cellular mechanisms driving organ precursor formation are often not clear. Here, using zebrafish, we investigate the epithelial rearrangements responsible for the development of the hemispherical retinal neuroepithelium (RNE), a part of the optic cup. We show that in addition to basal shrinkage of RNE cells, active migration of connected epithelial cells into the RNE is a crucial player in its formation. This cellular movement is driven by progressive cell-matrix contacts and actively translocates prospective RNE cells to their correct location before they adopt neuroepithelial fate. Failure of this migration during neuroepithelium formation leads to ectopic determination of RNE cells and consequently impairs optic cup formation. Overall, this study illustrates how spatiotemporal coordination between morphogenic movements and fate determination critically influences organogenesis.

*For correspondence: norden@mpi-cbg.de

**Competing interests:** The authors declare that no competing interests exist.

## Introduction

In many developmental contexts, organ formation includes rearrangements of epithelial sheets. Such rearrangements give rise to organ precursors that later develop into mature organs. For example, epithelial rearrangements that form the imaginal discs of *Drosophila* larvae generate organs in the adult fly including wings and legs (*Morata, 2001*). Similarly, the vertebrate neural tube is shaped by epithelial reorganization and later develops into the brain and the spinal cord (*Greene and Copp, 2014*). Epithelial reorganization occurs via changes in the morphology, number and location of cells, and ultimately defines the architecture of the developing organ (*Lecuit and Le Goff, 2007*). When epithelial reorganization and thereby organ precursor architecture is impaired, the structure and function of the mature organ can be compromised. For instance, defects in cell-matrix adhesion resulting in impaired wing imaginal disc formation ultimately cause a blistered wing (*Domínguez-Giménez et al., 2007*). Similarly, defects in epithelial fusion of neural folds can lead to problems in neural tube closure and generate severe birth defects in mammals (*Greene and Copp, 2014*). Hence, deciphering how epithelial morphogenesis shapes organ precursors is crucial to understand overall organ development.

One outstanding model to investigate how epithelial biology shapes organ architecture is the developing vertebrate retina. Here, the retinal neuroepithelium (RNE) is the organ precursor that later gives rise to all neurons of the mature retina (*Fuhrmann, 2010*). The hemispheric RNE that is located in the optic cup develops from the epithelial optic vesicles (*Bazin-Lopez et al., 2015*). Its formation involves complex epithelial rearrangements including tissue elongation, sheet invagination and epithelial sheet movements (*Martinez-Morales et al., 2009*; *Heermann et al., 2015*;

*Kwan et al., 2012*). It has been shown in mouse and human retinal organoid in vitro cultures that the optic vesicle epithelium self-organizes into a hemispherical shape due to high proliferation in a confined space (*Eiraku et al., 2011*; *Nakano et al., 2012*). However, work in zebrafish and *Xenopus* shows that RNE development continues even when cell proliferation is blocked (*Harris and Hartenstein, 1991*; *Kwan et al., 2012*). Such differences highlight the importance of in vivo studies of optic cup formation to address how the RNE is formed during embryonic development.

Due to its unmatched imaging potential, the zebrafish is an excellent model to understand in vivo optic cup formation at both the cellular and the tissue level. In teleosts, RNE morphogenesis occurs by rearrangements of a continuous epithelium, the bilayered optic vesicle (*Schmitt and Dowling, 1994*). The distal layer of the optic vesicle develops into the RNE and part of the proximal layer develops into retinal pigment epithelium (RPE). Work in zebrafish and medaka showed that basal constriction of RNE cells is important for RNE invagination (brown cell, *Figure 1A*) (*Martinez-Morales et al., 2009*; *Bogdanović et al., 2012*; *Nicolás-Pérez et al., 2016*). However, given that a subpopulation of prospective RNE cells is located in the proximal epithelial layer, at the onset of optic cup morphogenesis (OCM), it is not clear whether basal constrictions alone can drive RNE formation or whether these cells play an additional role. The proximal prospective RNE cells move into the distal, invaginating neuroepithelium by a process called rim involution (blue cell, *Figure 1A*) (*Kwan et al., 2012*; *Picker et al., 2009*; *Heermann et al., 2015*). However, to date, it remains unclear which molecular mechanisms drive rim involution and whether it is actively involved in RNE morphogenesis.

Here, we use a multi-scale approach to investigate these questions at the single-cell and the tissue level. We find that in addition to basal invagination of the RNE, rim involution critically supports RNE morphogenesis. Rim cells migrate actively and collectively to integrate in the invaginating RNE. When rim migration is perturbed, not all prospective neuroepithelial cells reach the RNE but nevertheless these cells adopt neuroepithelial fate. This results in severely disturbed retinal architecture. Thus, active migration of rim cells coordinates the timely integration of future neuroepithelial cells into the hemispherical RNE and is essential to prevent ectopic fate specification of neuroepithelial cells.

## Results

### Invagination of the retinal neuroepithelium involves basal accumulation of contractile actomyosin and basal cell surface reduction

To begin to elucidate the mechanisms of RNE formation, we initially focused on RNE invagination. It has been recently shown that RNE invagination is accompanied by basal constriction of neuroepithelial cells in the distal layer of the optic vesicle (*Figure 1A*, brown cell) (*Nicolás-Pérez et al., 2016*). To validate this finding, we labeled the cell cortex at different stages of optic cup morphogenesis from 15 somite stage (ss) to 28 ss (16 hours post fertilization (hpf) to 24 hpf) using the F-actin marker phalloidin. We measured the average cell density at the base of the RNE and noted that basal cell density increased significantly during optic cup invagination (*Figure 1—figure supplement 1A,C*). Concomitantly, the average basal area of RNE cells was reduced by about 45%. This reduction mainly occurred during the first half of the invagination process between 15 ss and 21 ss (*Figure 1B*). In contrast, the average apical area of cells increased. As a result, the average apical area was larger than the average basal area at the end of the process. This shows that cells undergo overall shape changes during invagination and form cones with a narrowing basal surface (*Figure 1B*).

To determine whether the observed reduction in basal cell area was linked to changes in actomyosin distribution, we imaged actin and myosin distribution during RNE morphogenesis. At the onset of neuroepithelial invagination, the actin marker GFP-UtrophinCH (*Burkel et al., 2007*) accumulated at the basal side of the RNE (*Video 1*, *Figure 1C*) similarly to the myosin marker, myl12.1-EGFP (*Iwasaki et al., 2001*) (*Video 2*, *Figure 1D*). Basal actin and myosin enrichment was specific to the invaginating zone of the neuroepithelium and was not observed in other regions of the optic cup (*Videos 1* and *2*, *Figure 1C,D*). These findings were corroborated by phalloidin (actin) and anti-phosphomyosin (active myosin) staining of samples fixed between 15 ss and 30 ss (*Figure 1E,G* and *Figure 1—figure supplement 1B*). Quantitative analysis of phalloidin and phosphomyosin intensity

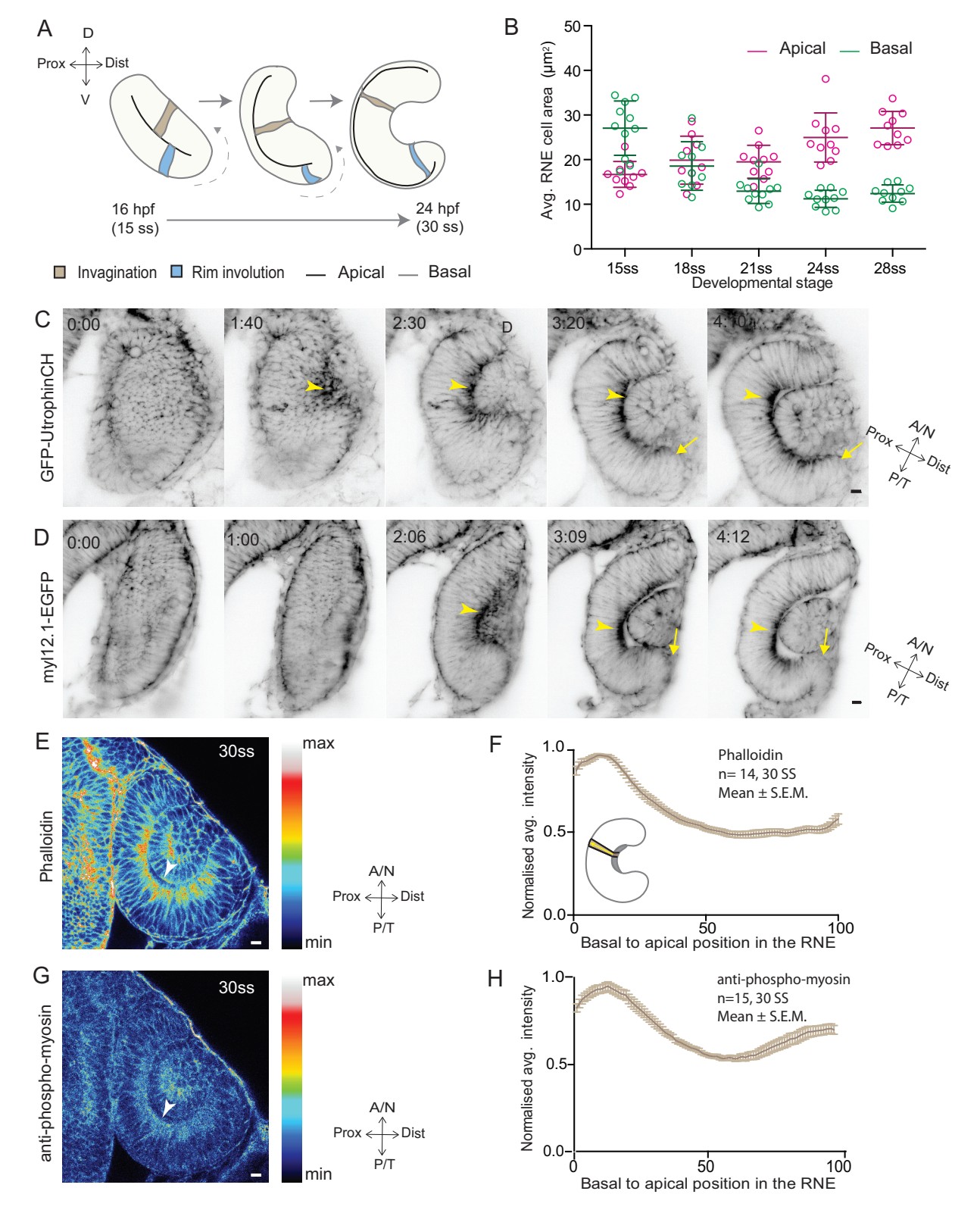

**Figure 1.** RNE invagination is accompanied by basal cell surface area shrinkage and basal actomyosin accumulation. (**A**) Schematic representation of RNE morphogenesis from 16 hours post fertilization (hpf) or 15 somite stage (ss) to 24 hpf or 30 ss showing RNE cells that undergo invagination (brown) and rim cells that undergo rim involution (blue). Dotted arrow marks direction of rim involution. (**B**) Average area of RNE cells at apical (magenta) and basal (green) sides during RNE morphogenesis with mean ± SD. N = 10 embryos. See *Figure 1—source data 1*. (**C,D**) Time-lapse imaging of RNE

*Figure 1 continued on next page*

*Figure 1 continued*

morphogenesis to assess the dynamics of actin marked by GFP-UtrophinCH (C) and of myosin marked by myl12.1-EGFP (D). Arrowhead marks basal actin and myosin enrichment in the RNE. Arrow marks the rim zone lacking the basal enrichment. Videos started around 15 ss. Time in h:min. Frames from *Videos 1* and *2*. (E,G) Confocal scan of optic cup at 30 ss immunostained for phalloidin (E) and phosphomyosin (G). The arrowhead marks enrichment at the basal side of the RNE. Lookup table indicates maximum and minimum intensity values. (F,H) Normalized average intensity distributions of phalloidin (F) and phosphomyosin (H) in the tissue volume along the apicobasal axis of the RNE at 30 ss. Mean ± SEM. The schematic shows a typical tissue section used for analysis (Also see *Figure 1—figure supplement 1D*). Tissue sections, n = 14 for phalloidin and n = 15 for phosphomyosin; N = 5 embryos. See *Figure 1—source data 2* and *3*. All scale bars = 10 µm. Developmental axes are shown next to the panels for orientation. D: Dorsal, V: Ventral; Prox: Proximal, Dist: Distal; A/N: Anterior or nasal, P/T: Posterior or temporal.

The following source data and figure supplement are available for figure 1:

**Source data 1.** Related to *Figure 1B*, *Figure 1—figure supplement 1C*.
**Source data 2.** Related to *Figure 1F*.
**Source data 3.** Related to *Figure 1H*.
**Figure supplement 1.** Analysis of the basal surface and actomyosin localisation during RNE invagination.

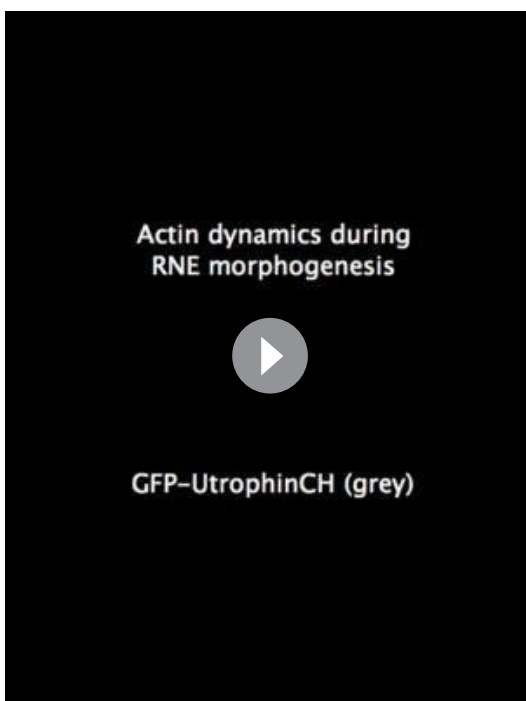
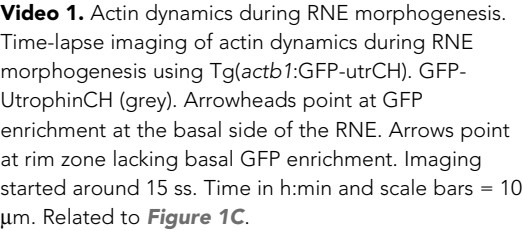

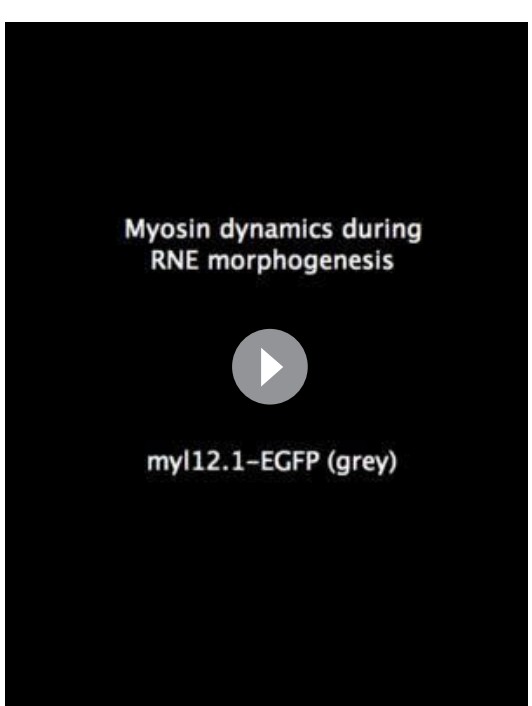

**Video 1.** Actin dynamics during RNE morphogenesis. Time-lapse imaging of actin dynamics during RNE morphogenesis using Tg(*actb1*:GFP-utrCH). GFP-UtrophinCH (grey). Arrowheads point at GFP enrichment at the basal side of the RNE. Arrows point at rim zone lacking basal GFP enrichment. Imaging started around 15 ss. Time in h:min and scale bars = 10 µm. Related to *Figure 1C*.

**Video 2.** Myosin dynamics during RNE morphogenesis. Time-lapse imaging of myosin dynamics during RNE morphogenesis using Tg(*actb1*:myl12.1-EGFP). Myl12.1-EGFP (grey). Arrowheads point at GFP enrichment at the basal side of the RNE. Arrows point at rim zone lacking basal GFP enrichment. Imaging started around 15 ss. Time in h:min and scale bars = 10 µm. Related to *Figure 1D*.

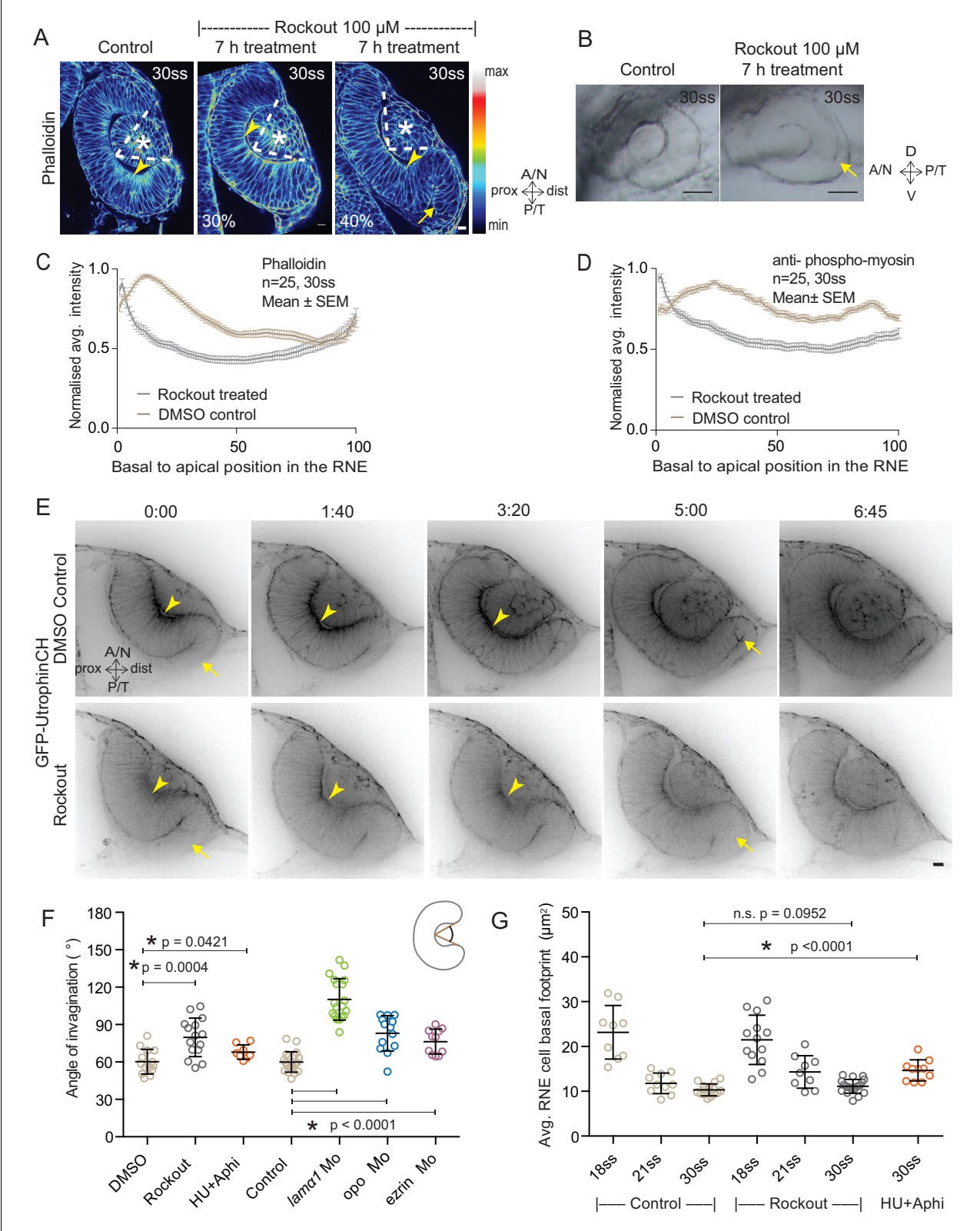

**Figure 2.** Impairment of actomyosin-driven basal constriction or proliferation does not prevent RNE formation. (**A**) Confocal scan of optic cup at 30 ss stained for phalloidin. Control (left), phenotypes after 7 h of Rockout treatment: invagination defect in 30% embryos (middle), invagination defect with epithelial accumulation in 40% embryos (right), (n = approx. 30 embryos, N = 5 experiments, see *Figure 2—source data 1*). Arrowheads mark the basal side of the RNE. Arrow marks the epithelial accumulation outside the RNE. Asterisk marks the developing lens. Dashed lines indicate the angle of

*Figure 2 continued on next page*

*Figure 2 continued*

invagination. Rockout treatment started around 13–14 ss. Lookup table indicates the minimum and maximum intensity values. Scale bar = 10 μm. (B) Brightfield images of optic cup at the end of the 7 h treatment. Control (left), treated with Rockout (right). Arrow marks the epithelial accumulation outside the RNE. Scale bar = 50 μm. (C,D) Normalized average intensity distributions of phalloidin (C) and phosphomyosin (D) in the tissue volume along the apicobasal axis of the RNE at 30 ss. Mean ± SEM. Control (brown), Rockout treated (grey). Tissue sections, n = 25; N = 5 embryos each. See *Figure 2—source data 2* and *3*. (E) Time-lapse of RNE morphogenesis in DMSO Control (upper) and Rockout-treated (lower) embryos expressing actin marker GFP-UtrophinCH. N = 5 embryos each. Rockout treatment was started 2 h before imaging around 13–14 ss. Imaging started at around 18 ss. Both movies were imaged simultaneously and under identical imaging conditions. Time in h:min. Frames from *Video 3*. Scale bar = 10 μm. (F) Invagination angle at 30 ss. Mean ± SD. The schematic shows the invagination angle as the angle held at the base of central RNE by the inner lips of the optic cup. P values for Mann-Whitney test with Control are as follows: Rockout p=0.0004, HU+Aphi p=0.0421 and for laminin a1 Mo, opo Mo and ezrin Mo p<0.0001. See *Figure 2—source data 4*. (G) Average basal area of RNE cells with mean ± SD. Each dot represents one embryo. P values for Mann-Whitney test with 30 ss control are as follows: 30 ss Rockout p=0.0952, 30 ss HU+Aphi p<0.0001. See *Figure 2—source data 5*.

The following source data and figure supplement are available for figure 2:

**Source data 1.** Related to *Figure 2A*, referred in text.
**Source data 2.** Related to *Figure 2C*.
**Source data 3.** Related to *Figure 2D*.
**Source data 4.** Related to *Figure 2F*.
**Source data 5.** Related to *Figure 2G*.
**Figure supplement 1.** Effect of Rockout and HU+Aphi treatment on the RNE.

distribution along the apicobasal axis of the neuroepithelium showed that the basal actomyosin bias spanned about 15% of the RNE height (*Figure 1F,H* and *Figure 1—figure supplement 1D*). Thus, we confirmed that invagination of the neuroepithelium is accompanied by basal surface area reduction (*Nicolás-Pérez et al., 2016*). We further show that invaginating RNE cells change their overall shape and feature basal accumulation of contractile actomyosin (*Figure 1F and H*).

## Inhibition of basal actomyosin contraction delays but does not prevent RNE invagination

To test whether the pool of contractile basal actomyosin is actively involved in RNE invagination, we perturbed myosin contractility using the drug Rockout, which is known to inhibit the Rock pathway upstream of actomyosin activity (*Yarrow et al., 2005*). Upon 7 h of Rockout treatment, lens structures in treated embryos developed similarly to controls (*Figure 2A*). In contrast, the basal actomyosin bias in invaginating RNE cells was markedly reduced (*Figure 2A,C,D*). Also, the overall staining for phosphomyosin was weaker in the treated embryos (*Figure 2—figure supplement 1A*). Despite this reduction of actin and myosin however, the effect on the morphology of the cup was surprisingly weak. 40% of embryos showed an accumulation of cells at the rim of the cup, whereas 30% of embryos showed a milder optic cup phenotype (n = 30 embryos per experiment, N = 5 experiments; 30% of embryos showed stalled development and were thus excluded from the analysis; *Figure 2A,B*, see *Figure 2—source data 1*). The invagination angle (defined as the angle between the inner lips of the optic cup and the center of the RNE, see

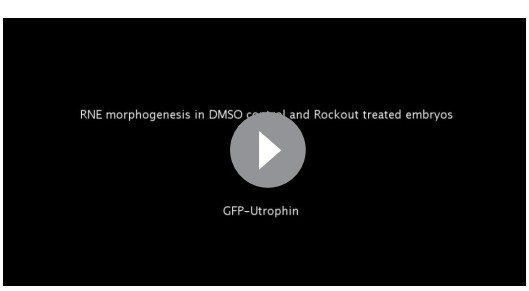

**Video 3.** RNE morphogenesis in Rockout treated embryos. Time-lapse imaging of Tg(*actb1*:GFP-utrCH) embryos treated with DMSO (left) and 125 μm Rockout (right). Treatment started 2 h before imaging at 13–14 ss. Imaging started around 18 ss. Time in h:min and scale bar = 10 μm. Related to *Figure 2E*.

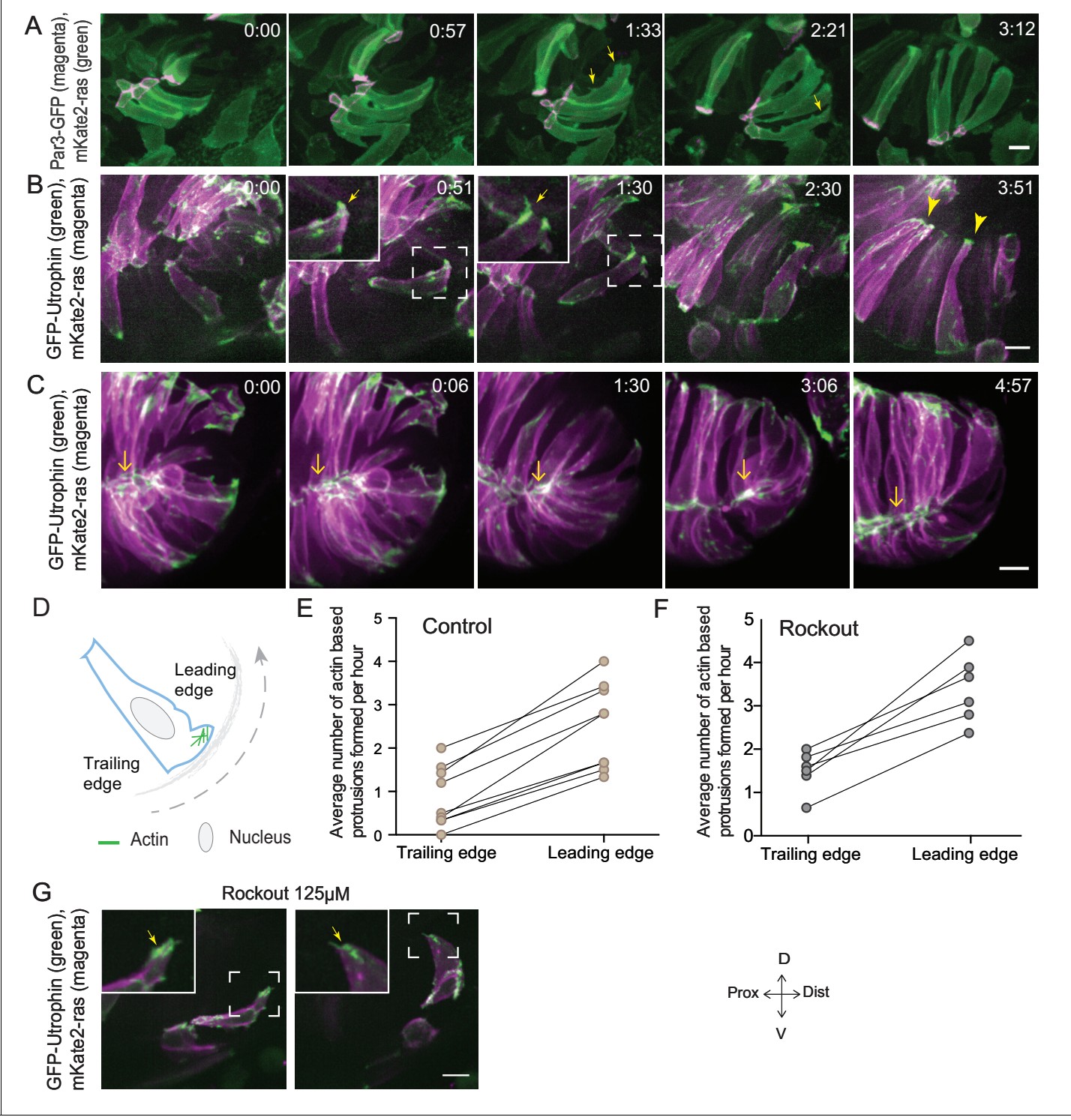

**Figure 3.** Rim involution involves active cell migration of connected epithelial cells. (**A**) Time-lapse imaging of rim zone with mosaic expression of Par3-GFP and mKate2-ras. Arrows show membrane protrusions. Frames from *Video 4*. N = 5. (**B**) Time-lapse imaging of rim zone with mosaic expression of GFP-UtrophinCH and ras-mKate2. Inlays show zoomed marked area. Arrows show actin localization in the protrusions. Arrowheads mark the basally enriched stable actin pool in the RNE. Frames from *Video 5*. N = 6. (**C**) Time-lapse imaging of rim zone with mosaic expression of GFP-UtrophinCH and ras-mKate2. Yellow arrows show apical actin localization at adherens junctions. Frames from *Video 7*. N = 6. (**D**) Schematic of a rim cell exhibiting actin based protrusions at the basal side. The arrow marks the direction of involution. The leading and lagging edges indicate the sides referred in (**E**) and (**F**). (**E**) Number of actin protrusions observed per hour in rim cells in the control condition. Each pair of datapoints represents two sides of the

*Figure 3 continued on next page*

*Figure 3 continued*

same rim cell. n = 9, N = 6 . See *Figure 3—source data 1*. (F) Number of actin protrusions observed per hour in the rim cells in Rockout treatment condition. Each pair of datapoints represents two sides of the same rim cell. n = 6, N = 6. See *Figure 3—source data 1*. (G) Confocal scan of rim zone in Rockout-treated embryos with mosaic expression of GFP-UtrophinCH and ras-mKate2. Inlays show enlarged marked area. Arrows show actin localization in the protrusions. Frames from *Video 6*. N = 6 all scale bars = 10 μm. All movies started around 17 ss -18 ss, Time in h:min.

The following source data and figure supplement are available for figure 3:

**Source data 1.** Related to *Figure 3E,F*.

**Figure supplement 1.** Microtubule and myosin dynamics during rim involution.

---

schematic in *Figure 2F* and Material and methods) was only moderately wider in Rockout treatement with an average of 80° (SEM ± 3.979 n = 15) compared to 60° (SEM ± 2.631 n = 14) in DMSO controls (*Figure 2F*). Thus, RNE invagination was initiated but was not as efficient in the treated embryos as in controls and appeared delayed (*Figure 2A,F*). To better understand the progression of RNE formation in the Rockout condition, we performed time-lapse imaging of GFP-UtrophinCH in Rockout-treated embryos. This confirmed that RNE invagination still took place, albeit with slower kinetics, even when basal actin levels were greatly reduced (*Video 3*, *Figure 2E,G*). Interestingly, at 30ss, the end point of optic cup invagination, the basal surface area of the invaginating cells was similar in controls and Rockout-treated embryos (*Figure 2G*). This observation suggests that RNE invagination does not solely depend on basal actomyosin contractility and that it can be compensated by other mechanisms.

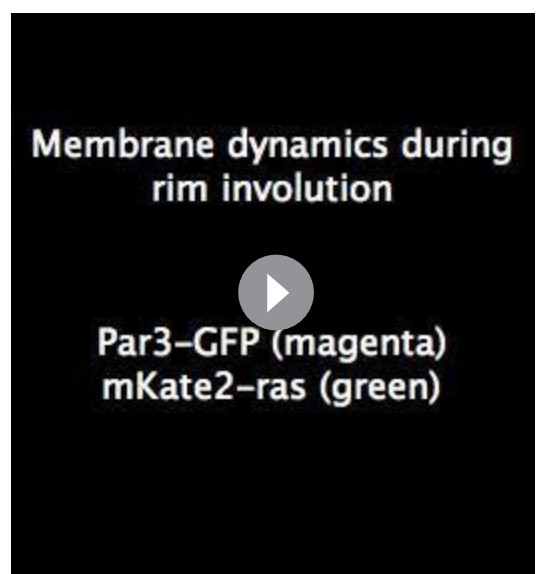

**Video 4.** Membrane dynamics during rim involution. Time-lapse imaging of membrane dynamics during rim involution. Mosaic expression of Par3-GFP (magenta) and mKate2-ras (green). Arrows point at the membrane protrusions. Imaging started at 17 ss-18 ss. Time in h:min and scale bar = 10 μm. Related to *Figure 3A*.

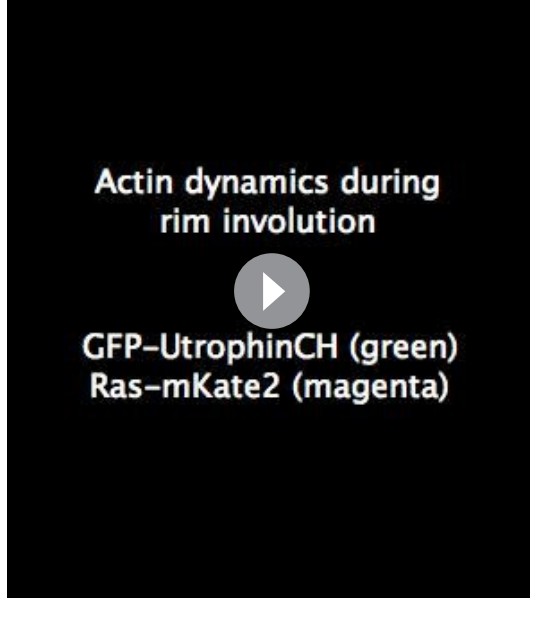

**Video 5.** Actin dynamics during rim involution. Time-lapse imaging of actin dynamics during rim involution. Mosaic expression of GFP-UtrophinCH (green) and mKate2-ras (magenta). Arrows point at utrophin localization in the membrane protrusions. Arrowheads point at the basal utrophin enrichment. Imaging started at 17 ss-18 ss. Time in h:min and scale bar = 10 μm. Related to *Figure 3B*.

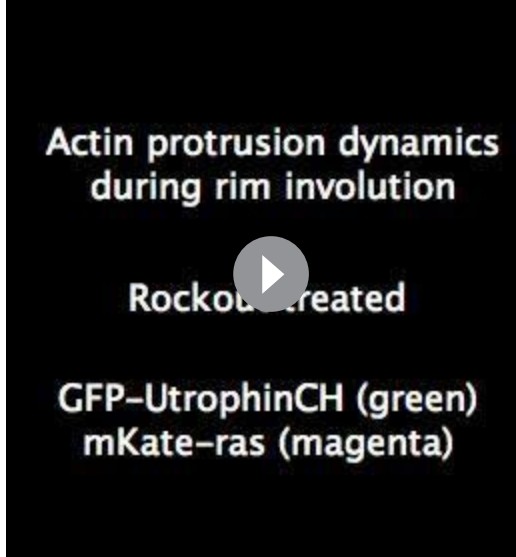

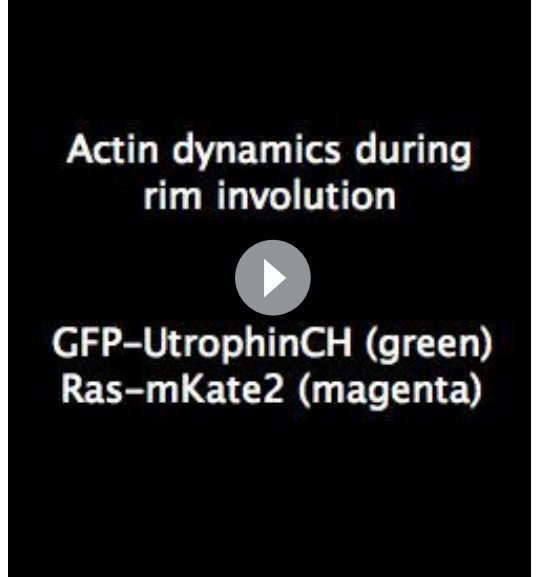

**Video 6.** Rim cell dynamics in Rockout treated embryos. Time-lapse imaging of actin dynamics in rim cells of Rockout treated embryos. Mosaic expression of GFP-UtrophinCH (green) and mKate2-ras (magenta). Rockout treatment started 2 h before imaging at 13–14 ss. Imaging started at 17 ss-18 ss. Time in h:min and scale bar = 10 µm. Related to *Figure 3G*.

**Video 7.** Actin dynamics during rim involution. Time-lapse imaging of actin dynamics during rim involution. Mosaic expression of GFP-UtrophinCH (green) and mKate2-ras (magenta). Arrows point at apical utrophin localization at the adherens junctions. Imaging started at 17 ss-18 ss. Time in h:min and scale bar = 10 µm. Related to *Figure 3C*.

## RNE morphogenesis persists in the absence of cell proliferation

RNE invagination could also be aided by cell compaction when an increasing number of cells inhabit the RNE and these cells contribute to bending of the basal surface. One way to increase cell numbers is proliferation, and it has been shown that RNE formation is accompanied by high proliferation rates (*Kwan et al., 2012*). In addition, proliferation has been suggested as a driving force for optic cup formation in vitro (*Eiraku et al., 2011*). In zebrafish and *Xenopus*, however, optic cup invagination persists even when proliferation is inhibited (*Kwan et al., 2012*; *Harris and Hartenstein, 1991*). To test whether tissue compaction caused by ongoing proliferation can at all influence RNE invagination, we inhibited proliferation with hydroxyurea and aphidicolin from 10 ss onwards (*Kwan et al., 2012*), leading to markedly reduced pH3 staining compared to controls (*Figure 2—figure supplement 1B*). Consequently, the neuroepithelium hosted fewer cells and these cells adopted a more columnar morphology than control cells (*Figure 2—figure supplement 1C,D* and [*Kwan et al., 2012*]). Furthermore, following inhibition of proliferation, cells in the RNE displayed a larger average basal area than control cells (*Figure 2G*). Despite this, optic cup formation still occurred and invagination angles were not severely affected (*Figure 2F* and *Figure 2—figure supplement 1C*). This observation indicated two things a) That the total number of cells in the invaginating RNE influences the basal area of single cells and b) that RNE invagination can take place even when cells show a broader basal area due to reduced cell proliferation. Thus, it suggests that other additional processes participate in RNE morphogenesis.

## RNE morphogenesis involves active cell migration at the rim of the optic cup

In addition to proliferation, another process that occurs concomitantly with RNE invagination that increases the cell number in the RNE is rim cell involution. This phenomenon is most prominent at the ventral and temporal side of the cup and relocates a substantial number of cells from the proximal layer of the optic vesicle into the invaginating RNE (blue cell, *Figure 1A*) (*Kwan et al., 2012*;

**Video 8.** Apical domain dynamics during rim involution. Time-lapse imaging of apical domain dynamics in embryos expressing of Par3-GFP (magenta) and mKate2-ras (green). The yellow box points at the zoomed area. Arrows point at the apical Par3 domain. Imaging started at 17 ss-18 ss. Time in h: min and scale bar = 10 μm.

**Video 9.** Myosin dynamics during rim involution. Time-lapse imaging of myosin dynamics during rim involution. Mosaic expression of DD-myl12b-GFP (green) and mKate2-ras (magenta). Arrows point at dynamic basal spots. Arrowheads point at the stable basal enrichment. Imaging started at 17 ss-18 ss. Time in h:min and scale bar = 10 μm. Related to *Figure 3—figure supplement 1B*.

**Video 10.** Integrin dynamics during rim involution. Time-lapse imaging of integrin dynamics during rim involution. Mosaic expression of Integrin*β*1b-mKate2 (magenta) and GFP-ras (green). Arrows point at the short-lived foci. Arrowheads point at the stable basal enrichment indicative of stable focal adhesions. Imaging started at 17 ss-18 ss. Time in h:min and scale bar = 10 μm. Related to *Figure 4A*.

**Video 11.** Paxillin dynamics during rim involution. Time-lapse imaging of paxillin dynamics during rim involution. Mosaic expression of paxillin-mKate2. Arrows point at the short-lived foci. Arrowheads point at the stable basal enrichment indicative of stable focal adhesions. Imaging started at 17 ss-18 ss. Time in h: min and scale bar = 10 μm. Related to *Figure 4B*.

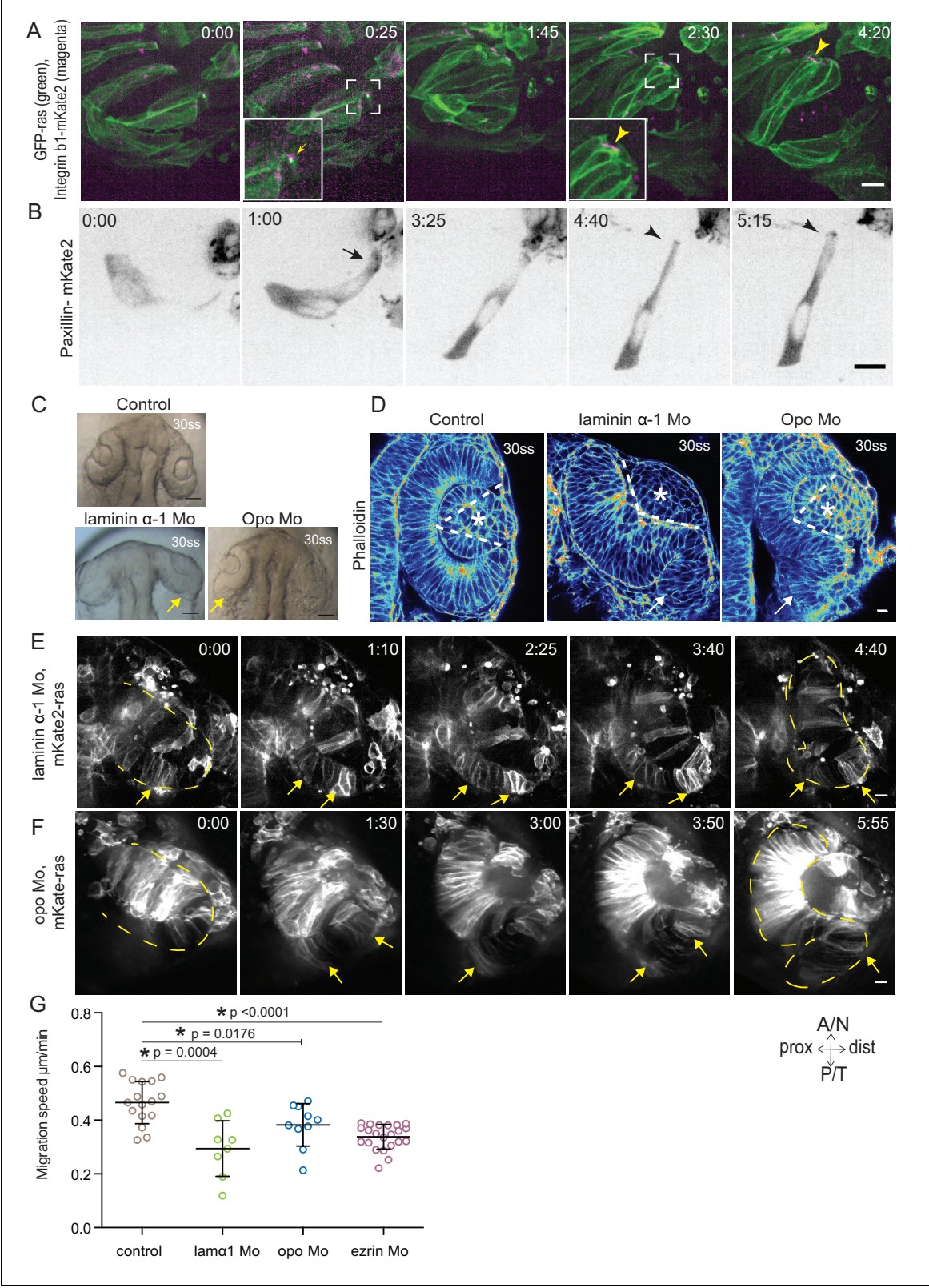

**Figure 4.** Dynamic cell-ECM attachment of rim cells is important for RNE morphogenesis. (**A**) Time-lapse imaging of rim zone with mosaic expression of GFP-ras and Integrin-β1b-mKate2. Inlays show enlarged marked area. Arrow indicates the integrin foci during migration. Arrowheads mark the basally enriched stable integrin pool in the RNE cell. Frames from *Video 10*. N = 4. Scale bar = 10 μm. Imaging started at 17–18 ss, Time in h:min. (**B**) Timelapse imaging of rim zone with mosaic expression of paxillin-mKate2. Arrow indicates the short-lived paxillin foci. Arrowhead marks the stable

*Figure 4 continued on next page*

*Figure 4 continued*

basal paxillin localization in the RNE. Frames from *Video 11*. N = 6. Scale bar = 10 µm. Imaging started at 17–18 ss. Time in h:min. (C) Brightfield images of the dorsal view of 30 ss embryo head. Control (upper), laminin morphant (lower left) and opo morphant (lower right). Arrows mark the epithelial accumulation outside the RNE. Scale bar = 50 µm. (D) Confocal scan of optic cups at 30 ss stained for phalloidin. Control (left), laminin morphant (middle) and opo morphant (right). Arrows mark the epithelial accumulation outside the RNE. Dashed lines indicate the angle of invagination. Asterisk marks the developing lens. Lookup table indicates the minimum and maximum intensity values. Scale bar = 10 µm. (E,F) Time-lapse imaging of RNE morphogenesis in laminin morphant (E) and Opo morphant (F) injected mosaically with mKate2-ras RNA. Arrows mark rim cells that failed to move. Dashed line marks the outline of the developing RNE. Frames from *Videos 13* and *14*. Time in h:min. Scale bar = 10 µm. Movies started at 16 ss-17 ss. (G) Migration speed of rim cells (Mean ± SD). P values for Mann Whitney test with control: laminin Mo p=0.0004, opo Mo p=0.0176, ezrin Mo p<0.0001. See *Figure 4—source data 2*.

The following source data and figure supplements are available for figure 4:

**Source data 1.** Related to *Figure 4—figure supplement 2D*.

**Source data 2.** Related to *Figure 4G*.

**Source data 3.** Related to *Figure 4—figure supplement 3C*.

**Source data 4.** Related to *Figure 4—figure supplement 3D*.

**Figure supplement 1.** Dynamics of ECM distribution and cell-ECM attachment during RNE morphogenesis.

**Figure supplement 2.** Evaluation of the efficiency of morpholino mediated knockdown of laminin α-1and opo.

**Figure supplement 3.** Effect of perturbed cell-ECM attachment on the optic cup.

*Picker et al., 2009*; *Heermann et al., 2015*). Such an influx of cells may result in increased tissue compaction and contribute to the inward bending of the tissue. As we noted that rim cells in the Rockout-treated embryos continued to move into the cup (*Figure 2E*, *Video 3*), we wondered whether these cell movements could explain the recovery of RNE morphogenesis defects upon Rockout treatment. We thus asked if rim involution was actively involved in shaping the RNE. To answer this question, we first needed to understand how cells move at the rim. Hence, we characterized their single-cell dynamics. Mosaic labeling of rim cells with the membrane marker mKate2-ras and the apical marker Par3-GFP revealed that moving rim cells kept a stable apical domain but were very dynamic at the basal side (*Figure 3A*, *Video 4*). As microtubules marked with EMTB-tdTomato did not show significant reorganization during rim involution, we concluded that microtubules were not majorly involved in the cellular dynamics (*Figure 3—figure supplement 1A*). Instead, we observed that cells extended dynamic lamellipodia-like protrusions in the direction of movement (*Figure 3A*, *Video 4*). These protrusions were actin rich, as shown by GFP-UtrophinCH localization (*Figure 3B*, *Video 5*), underlining their lamellipodial character. Such lamellipodia were only observed in the rim cells and were not seen in the invaginating RNE cells (*Figure 3B*, *Video 5*). When we quantified basal protrusive activity during rim migration, we found that the vast majority of the actin rich protrusions occurred in the direction of movement, the leading side of the cells facing the invaginating zone (*Figure 3D,E*). This showed that the protrusive activity of rim cells is directional. Interestingly, the directional actin-rich lamellipodia formation and rim migration were still observed in the Rockout condition (*Figure 3F,G*, *Video 6*), further indicating that the migratory behavior of rim cells occurs independently of the basal actomyosin bias in the invaginating RNE cells. Interestingly, while rim cells exhibited protrusive membrane activity at the basal side, they nevertheless maintained their apicobasal polarity and an apical actin pool associated with the adherens junctions (*Figure 3C*, *Videos 7* and *8*). This shows that a) rim cells are integrated in an intact epithelial sheet and display the migratory behavior as a collective epithelium and b) that rim cells show characteristics that are intermediate between the epithelial and mesenchymal state.

However, once rim cells reached the RNE, basal actin dynamics changed considerably and mesenchymal characteristics were lost. Lamellipodia formation ceased and a pool of stable basal actin was observed similar to the actin distribution in the invaginating RNE cells (*Figure 3B*, *Video 5*).

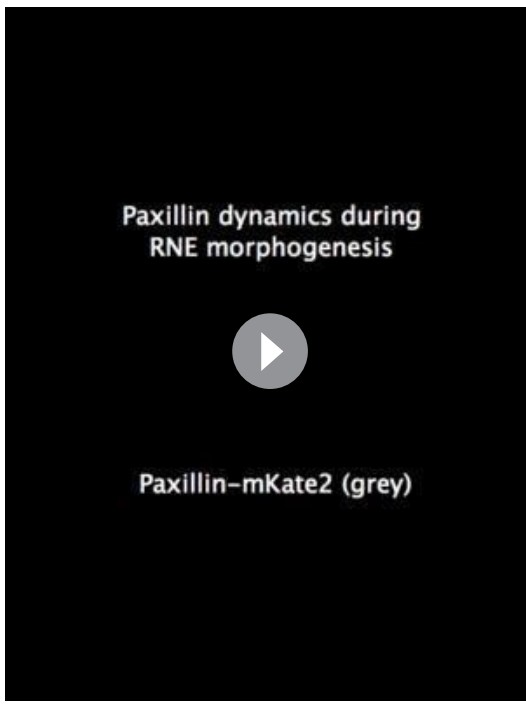

**Video 12.** Paxillin dynamics during RNE morphogenesis. Time-lapse imaging of paxillin dynamics during RNE morphogenesis in paxillin-mKate2 RNA injected embryos. Paxillin-mKate2 (grey). Imaging started around 15 ss. Time in h:min and scale

Likewise, while myosin labeled by DD-myl12b-GFP occurred as dynamic spots at the basal side of cells during rim movement (*Video 9*, *Figure 3—figure supplement 1B*), it became stably enriched once the cells reached the inner side of the optic cup. Together, these data show that rim cells exhibit dynamic and directional basal lamellipodia formation that coincides with directed migratory behavior. Once cells reach the RNE, however, they change their basal dynamics and establish a stable basal actomyosin pool similar to the basally constricting RNE cells.

## Rim cell migration depends on cell-ECM adhesion

Our analysis so far suggested that while being integrated in an epithelium, rim cells exhibit migratory behavior as a collective driven by directed basal lamellipodial activity. We next asked how exactly lamellipodia formation can drive rim cell movement. Lamellipodia-driven migration often depends on dynamic attachments of cells to the surrounding extracellular matrix (ECM) (*Friedl and Wolf, 2010*). Thus, we assessed the distribution of ECM during rim movement by immunostainings for the ECM components Laminin, Chondroitin sulfate and

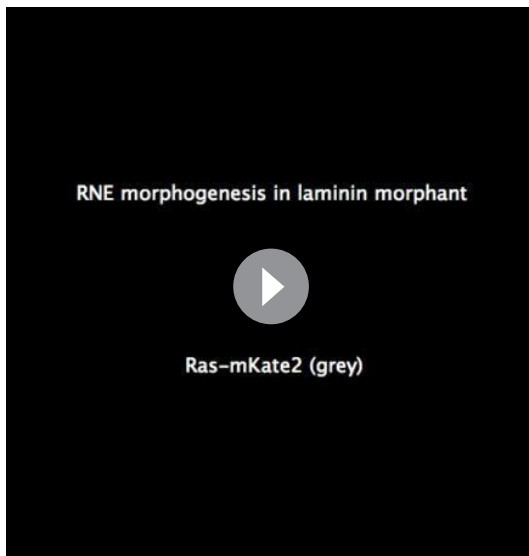

**Video 13.** RNE morphogenesis in laminin morphant. Time-lapse imaging of mosaic expression of mKate2-ras in laminin morphant. Lines mark the outline of developing optic vesicle and later RNE. Arrows point at rim cells with perturbed rim involution. Imaging started 16 ss-17 ss. Time in h:min and scale bar = 10 μm. Related to *Figure 4E*.

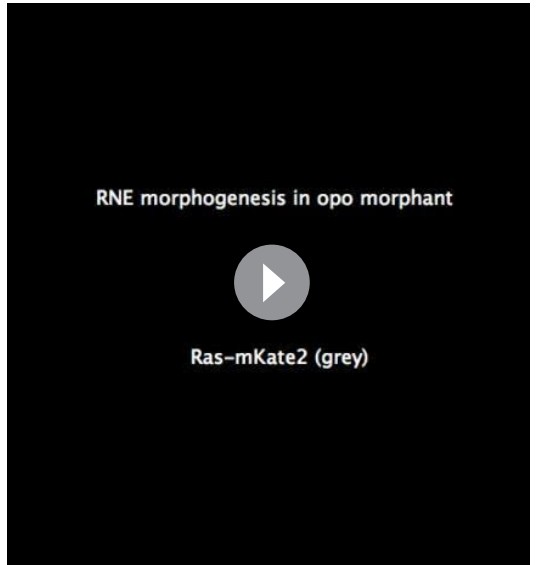

**Video 14.** RNE morphogenesis in Opo morphant. Time-lapse imaging of mosaic expression of mKate2-ras in opo morphant. Lines mark the outline of developing optic vesicle and later RNE. Arrows point at rim cells with perturbed rim involution. Imaging started 16 ss-17 ss. Time in h:min and scale bar = 10 μm. Related to *Figure 4F*.

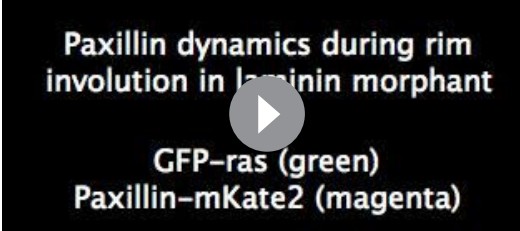

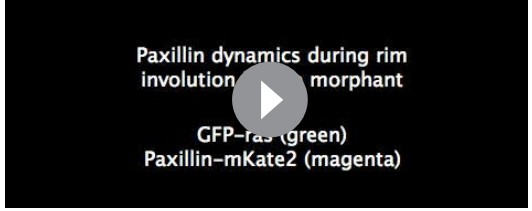

**Video 15.** Paxillin dynamics during rim involution in laminin morphant. Time-lapse imaging of paxillin dynamics during rim involution in laminin morphant with mosaic expression of GFP-ras (left) and paxillin-mKate2 (middle). Red arrowhead points at the stable paxillin localisation. Blue arrowheads point at the basal blebbing in the rim cells. Time in h:min and scale bar = 10 μm. Related to *Figure 5—figure supplement 1A*.

**Video 16.** Paxillin dynamics during rim involution in opo morphant. Time-lapse imaging of paxillin dynamics during rim involution in opo morphant with mosaic expression of GFP-ras (left) and paxillin-mKate2 (middle). Red arrowhead points at paxillin foci. Blue arrowheads point at the basal blebbing in the rim cells. Time in h:min and scale bar = 10 μm. Related to *Figure 5—figure supplement 1B*.

Fibronectin. Laminin and Chondroitin sulfate were only weakly expressed beneath the developing retinal pigment epithelium (RPE), but showed stronger accumulation in the rim zone, the developing lens placode and the invaginating RNE (*Figure 4—figure supplement 1A*). Fibronectin instead showed preferential staining in the rim zone and the developing lens placode and weaker staining beneath the invaginating RNE (*Figure 4—figure supplement 1B*). Hence, we speculated that the migrating rim cells use their lamellipodia to dynamically attach to the underlying ECM and thereby generate force for movement. Imaging the focal adhesion markers Integrin-$\beta$1-mKate2 and Paxillin-mKate2 indeed revealed dynamic short-lived foci at the basal side of rim cells during time-lapse imaging (*Videos 10* and *11*, *Figure 4A,B*). However, once cells reach the RNE and stopped the active migratory behavior they formed stable focal adhesions marked by integrin and paxillin as well as actin (*Videos 10* and *11*, *Figure 4A,B*). Co-imaging of the localization and dynamics of Paxillin-mKate2 and GFP-UtrophinCH corroborated this finding. While the cells in the rim zone lacked basal enrichment of actin and paxillin, cells in the invaginating region showed stable basal enrichment of Paxillin coinciding with the basal accumulation of actin (*Figure 4—figure supplement 1C*, *Video 12*).

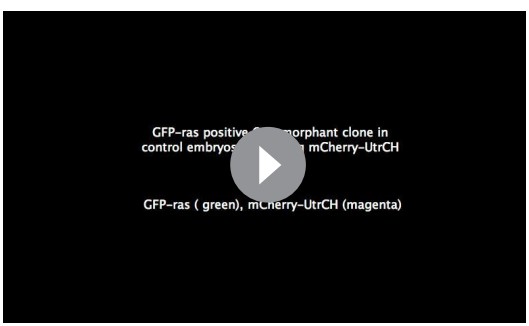

**Video 17.** GFP-ras expressing opo morphant clone in mCherry-UtrCH expressing control background. Time-lapse imaging of RNE morphogenesis in Tg(actb1: mCherry-UtrCH) acceptor embryo transplanted with GFP-ras positive opo morphant cells. Opo morphants cells (green), acceptor cells (magenta). Arrowheads point at the basal blebbing in the rim cells. Time in h: min and scale bar = 10 μm. Related to *Figure 5— figure supplement 1D*.

**Video 18.** Protrusion dynamics during rim involution in ezrin morphant. Time-lapse imaging of membrane protrusion dynamics during rim involution in ezrin morphant with mosaic expression of GFP-mKate2 (left) and GFP-UtrophinCH (middle). Blue arrowheads point at the basal blebbing in the rim cells. Time in h:min and scale bar = 10 μm

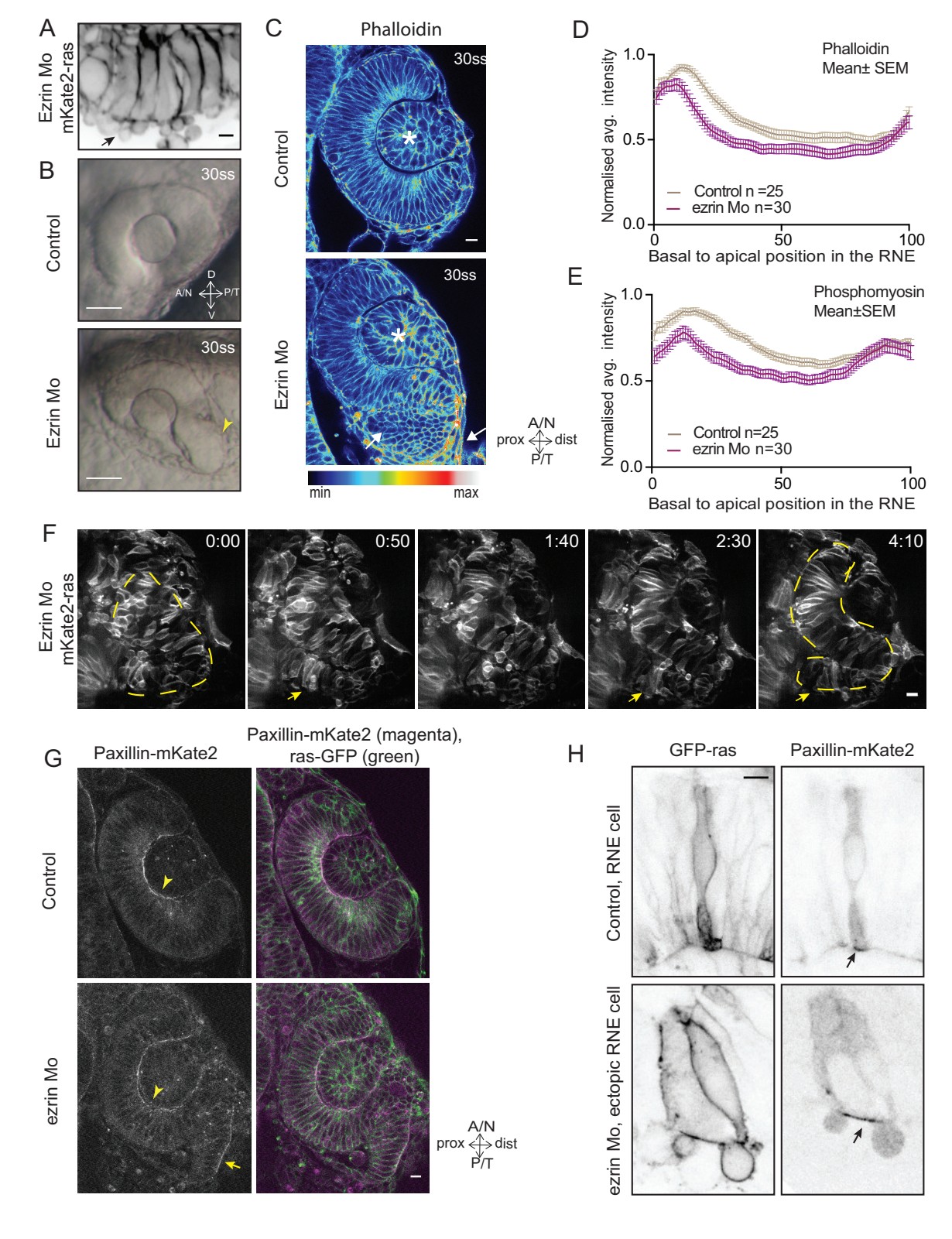

**Figure 5.** Perturbed basal lamellipodial activity leads to compromised rim involution and impairs RNE morphogenesis. (**A**) Confocal scan of rim cells in mKate2-ras injected ezrin morphant. Arrow indicates basal blebs in rim cells. Scale bar = 5 μm. (**B**) Brightfield images of side view of 30 ss optic cup. Control (top) and ezrin morphant (bottom). Arrowhead marks the epithelial accumulation outside the RNE. Scale bar = 50 μm. (**C**) Confocal scan of optic cup at 30 ss stained for phalloidin. Control (top) and ezrin morphants (bottom). Arrows mark the epithelial accumulation outside the RNE. Asterisk

*Figure 5 continued on next page*

*Figure 5 continued*

marks the lens. Lookup table indicates the minimum and maximum intensity values. Scale bar = 10 μm. (D,E) Normalized average intensity distributions of phalloidin (D) and phosphomyosin (E) in the tissue volume along the apicobasal axis of the RNE at 30 ss. Mean ± SEM. Control (brown) ezrin Mo (magenta). Tissue sections, n = 25 for control and n = 30 for ezrin Mo; N = 5 embryos each. See *Figure 5—source data 1*, *2*. (F) Time-lapse imaging of RNE morphogenesis in ezrin morphant injected mosaically with ras-mKate2 RNA. Arrows mark rim cells that failed to move. Dashed line marks the outline of developing RNE. Frames from *Video 18*. Time in h:min. Scale bar = 10 μm. Imaging started at 17 ss – 18 ss. (G) Confocal scan of 30 ss optic cup expressing paxillin-mKate2 and GFP-ras. Control (top), ezrin Mo (bottom). Arrow marks paxillin enrichment at the basal side. N = 7. Scale bar = 10 μm. (H) Confocal scan of paxillin-mKate2 and GFP-ras expressing cells. Control RNE cells (top), ectopic RNE cells in ezrin morphant (bottom). Arrow marks basal paxillin enrichment. Scale bar = 5 μm.

The following source data and figure supplements are available for figure 5:

**Source data 1.** Related to *Figure 5E*.
**Source data 2.** Related to *Figure 5—figure supplement 2E*
**Source data 3.** Related to *Figure 5D*.
**Figure supplement 1.** Effect of perturbed cell-ECM attachment on the rim cells.
**Figure supplement 2.** Analysis of ezrin morphant condition.

This suggested that the active migratory behavior observed for rim cells is confined to the zone that connects the developing RPE and the RNE, where cells move from one epithelial layer into the other by turning 180°. Once the cells reach the inner layer of the cup they change to stable cell-ECM attachment and attain typical neuroepithelial morphology.

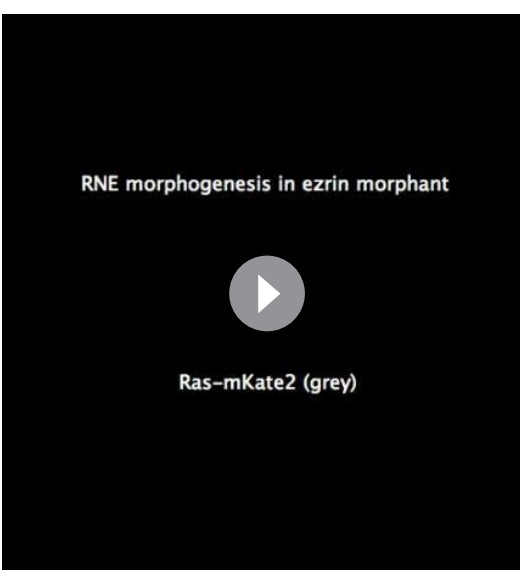

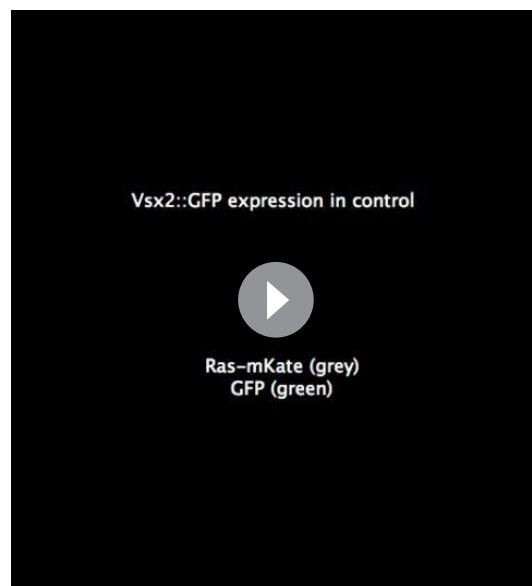

**Video 19.** RNE morphogenesis in ezrin morphant. Time-lapse imaging of mosaic expression of mKate2-ras in ezrin morphant. Lines mark the outline of developing optic vesicle and later RNE. Arrows point at rim cells with perturbed rim involution. Imaging started 16 ss-17 ss. Time in h:min and scale bar = 10 μm. Related to *Figure 5F*.

**Video 20.** *vsx2*::GFP expression during RNE morphogenesis. Time-lapse imaging of control RNE morphogenesis in Tg(*vsx2*::GFP, *βactin*::mKate2-ras). Arrow points at appearance of bright GFP signal indicative of Vsx2 expression. Imaging started 16 ss-17 ss. Time in h:min and scale bar = 10 μm. Related to *Figure 6A*.

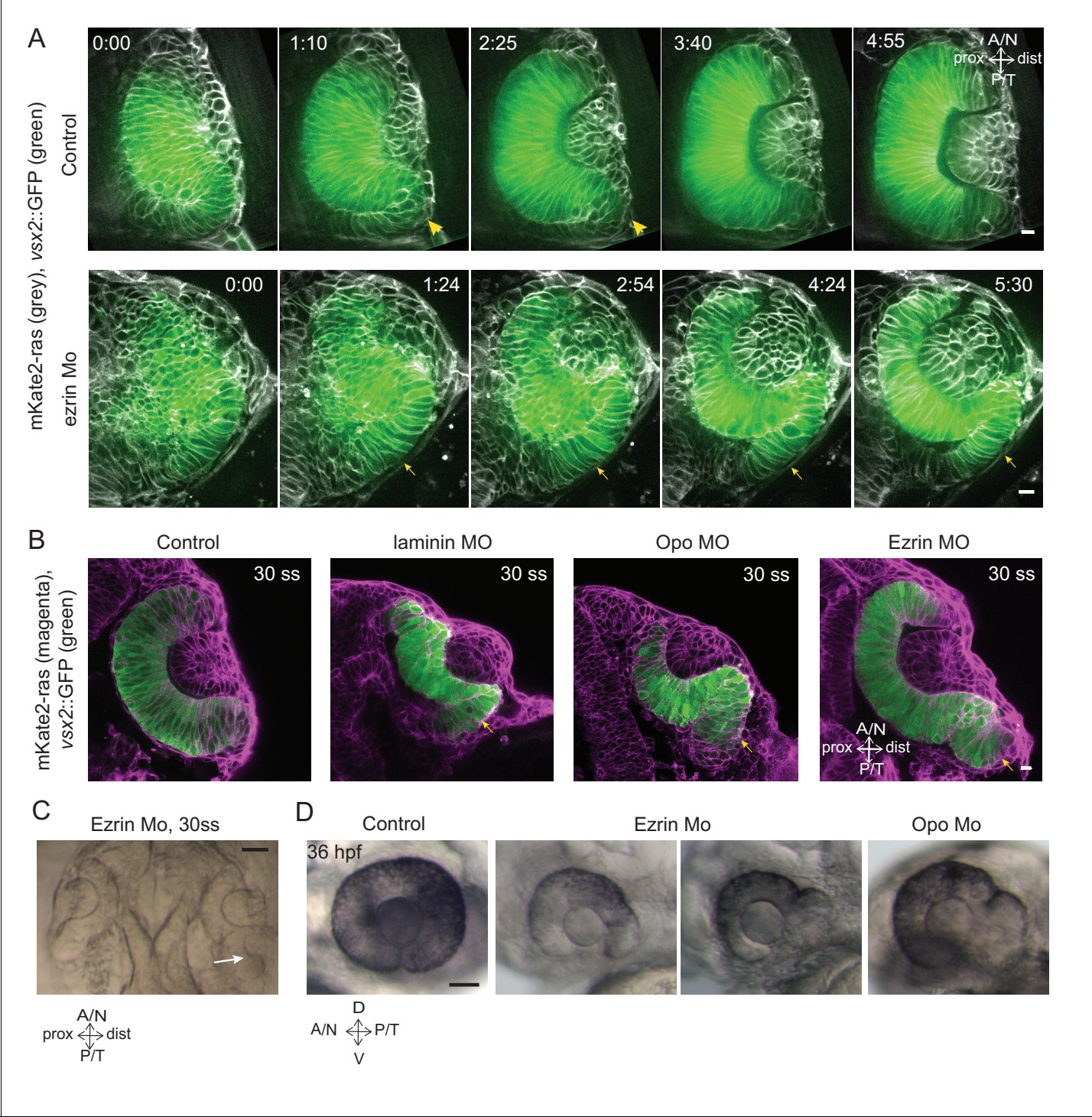

**Figure 6.** Impairment of rim involution leads to mispositioning of neuroepithelial cells. (**A**) Time-lapse imaging of control and *ezrin* morphant condition in Tg(*vsx2*::GFP, *βactin*::mKate2ras) background. Arrowheads mark rim zone in control. Arrows mark rim cells that failed to migrate in *ezrin* morphant. Frames from *Videos 19* and *20*. Time in h:min. Videos started at 16 ss -17 ss. N = 9. (**B**) Confocal scans of optic cups at 30 ss in control, laminin morphant, Opo morphant and ezrin morphant conditions in Tg(*vsx2*::GFP, *βactin*::mKate2-ras). Embryos were stained for GFP and mKate-2. Arrows point at the ectopic RNE cells. N = 7 each. (**C**) Brightfield image of 30 ss *ezrin* morphant showing secondary optic cup phenotype. Arrow marks secondary invagination zone. (**D**) Brightfield images of 36 hpf control embryos, ezrin morphants and opo morphant. Scale bars in (**A**,**B**) =10 μm and (**C**, **D**) = 50 μm.

To verify that such dynamic interactions with the ECM were indeed necessary for successful rim cell migration, we perturbed cell-matrix attachment by interfering with the ECM or the intracellular attachment side. To this end, we knocked down laminin α-1 and ojoplano/opo using previously published morpholinos (*Martinez-Morales et al., 2009*; *Pollard et al., 2006*) (see *Figure 4—figure supplement 2A–D* for controls for efficiency of morpholino knockdown). Laminin depletion interferes directly with ECM integrity and it has been shown to be involved at different stages of optic cup development (*Ivanovitch et al., 2013*; *Bryan et al., 2016*). We observed that depletion of laminin from the ECM resulted in an overall increase in Fibronectin staining, indicating a change in the ECM composition and arrangement (*Figure 4—figure supplement 3A*). Knockdown of laminin resulted in impaired optic cup formation at 30 ss. Epithelial tissue accumulated at the temporal side of the optic cup was observed, giving the cup an S-shaped appearance instead of the C-shaped appearance seen in controls (*Figure 4C,D*).

To test if intracellular perturbation of cell-ECM attachment would also affect rim migration, we knocked down Opo. Opo is a protein that is known to regulate endocytosis of integrins and was previously shown to play an important role in eye morphogenesis (*Martinez-Morales et al., 2009*; *Bogdanović et al., 2012*). Indeed, opo morphants recapitulated the S-shaped optic cup phenotype, although the severity of the phenotype was milder than in the laminin condition (*Figure 4C,D*).

Time-lapse imaging of both conditions confirmed that the S-shaped phenotype was caused by impaired rim cell migration, as the speed of rim cells was reduced (*Videos 13* and *14*, *Figure 4E,F, G*). This reduced speed most likely results from the inability of migrating cells to efficiently use their lamellipodia to migrate along the underlying matrix. A recent study reported that Laminin α-1 mutant shows polarity defects in cells of the optic cup (*Bryan et al., 2016*). However, in the morphant condition used here we did not observe a strong effect on polarity. We only observed few ectopic apical domains within the neuroepithelium while overall tissue-wide apical marker distribution was not affected in the laminin morphants (*Figure 4—figure supplement 3B*). Thus, this condition allowed us to dissect the effects of perturbed cell-ECM adhesion during OCM. Similarly, no defects on apical marker distribution were observed in Opo morphants (*Figure 4—figure supplement 3B*, *Martinez-Morales et al., 2009*). Basally however, we noticed that actomyosin accumulation was reduced in the invagination zone in both laminin and opo knockdown conditions (*Figure 4D*, *Figure 4—figure supplement 3C,D*). In addition, invagination angles were affected (*Figure 2F*). This argued that both, rim migration and actomyosin-assisted invagination were disturbed in these conditions leading to the strong phenotypes observed.

Together, these data imply that interference with cell-ECM attachment leads to disturbance of the two cell behaviors involved in hemispheric RNE formation, rim cells cannot actively migrate due to impaired cell ECM-attachments and otherwise invaginating RNE cells failed to establish the basal actomyosin bias.

## Perturbed basal lamellipodial activity affects migration of rim cells and leads to defects in RNE architecture

To elucidate the cellular changes that led to the impaired rim cell migration upon interference with cell-ECM attachment, we analyzed the basal membrane dynamics of rim cells in the laminin and opo knockdown conditions. When we imaged the focal adhesion component Paxillin using mosaic Paxillin-mKate2 RNA injection, we

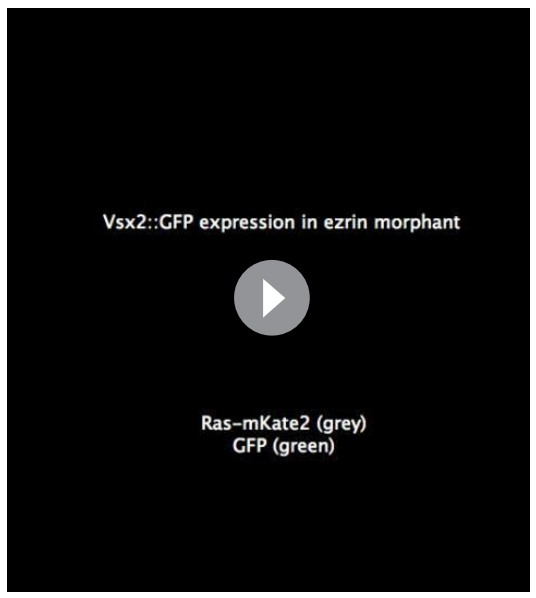

**Video 21.** *vsx2*::GFP expression during RNE morphogenesis in ezrin morphant. Time-lapse imaging of RNE morphogenesis in Tg(*vsx2*::GFP, *βactin*:: mKate2-ras) in ezrin morphant. Arrow points at appearance of bright GFP signal indicative of *vsx2* expression. Imaging started 16 ss-17 ss. Time in h:min and scale bar = 10 μm. Related to *Figure 6A*

observed that in laminin morphants the rim cells localized Paxillin stably along the modified basement membrane instead of the dynamic foci seen in the control condition (*Video 15*, *Figure 5—figure supplement 1A*). In opo morphants however, Paxillin distribution was more transient than seen in the controls (*Video 16*, *Figure 5—figure supplement 1B*). In both cases the rim cells exhibited basal bleb-like protrusions instead of the lamellipodia observed in control embryos (*Videos 15 and 16*, *Figure 5—figure supplement 1A,B*). These bleb-like protrusions were extension of the plasma membrane, with short lived actin localization and devoid of myosin, showing the characteristic features of blebs (*Figure 5—figure supplement 1C*). Such basal blebs were also observed when opo morphants cells were transplanted into control embryos, in line with a cell autonomous phenotype (*Video 17*, *Figure 5—figure supplement 1D*, N = 4 out of 5 transplanted embryos). The observation that morphant cells nevertheless reached the RNE underlines the collective characteristic of the migration as surrounding control cells most likely pull them along to their final positions (*Video 17*). The finding that the cells that failed to undergo active rim migration exhibited blebs instead of lamellipodia made us ask whether perturbation of the normal protrusive activity itself could interfere with rim cell migration. To test this idea, we knocked down the ERM protein family member Ezrin using a published morpholino (*Link et al., 2006*). Ezrin is a protein that links actin and the plasma membrane and its knockdown has been shown to lead to increased blebbing and reduced migration rates of zebrafish prechordal plate precursor cells (*Schepis et al., 2012*) (see *Figure 5—figure supplement 2A,B* for controls for efficiency of morpholino knockdown by Western blot and overall morphant morphology). During RNE formation, we observed that ezrin knockdown did not affect the overall apicobasal polarity in the developing optic cup (*Figure 5—figure supplement 2C*) but specifically induced basal blebbing in the area where rim cells resided, whereas lateral sides of the rim cells and the pool of invaginating RNE cells were not affected (*Video 18*, *Figure 5A*). We also observed rim cell blebbing when ezrin morphant cells were transplanted into control background, again arguing that blebbing is a cell autonomous phenotype ([*Figure 5—figure supplement 2D*] N = 2 out of 4 transplanted embryos. This variability is most likely due to varying morpholino concentration within transplanted cells as they come from different donors). Interestingly, ezrin depletion in all rim cells resulted in a phenotype similar to Laminin or Integrin interference. The average cell speed was reduced by 27% and rim cells did not reach the RNE (*Figures 4G* and *5B,C and F*, *Video 19*). While the formation of the lens and the RPE occurred similar to the control situation (*Figure 5C*), specifically RNE morphogenesis in ezrin morphants was defective and at 30 ss an S-shaped optic cup was observed (*Figure 5B,C*). Importantly, however, we only observed a minor effect on basal actomyosin accumulation in the RNE (*Figure 5D,E*). A comparison of the instantaneous slopes of the normalized phalloidin intensity distribution curves confirmed that unlike Rockout, laminin Mo and Opo Mo, ezrin morphants showed very little disturbance of the basal actomyosin (*Figure 5—figure supplement 2E*). This indicated that the S-shaped phenotype was mainly caused by the defects in rim involution and less by the perturbation of basal actomyosin. This S-shaped phenotype did not recover upon continued development (*Figure 5B*). In contrast, the cells that stayed outside the presumptive RNE layer eventually showed a stable Paxillin distribution outside the RNE as seen in the laminin knock down condition (*Figure 5G,H*). Cells thereby attained neuroepithelial-like morphology. We conclude that correct ECM formation and distribution is important for lamellipodia formation and successful rim cell migration. When cell-ECM attachment is impaired, lamellipodia formation is perturbed and rim cells form blebs but these blebs cannot support cellular movement. Consequently, cells do not enter the neuroepithelium but can nevertheless take on neuroepithelial morphology at ectopic location.

## Rim cell migration ensures timely entry of RNE cells into the optic cup

As interference with rim movement led to an accumulation of cells outside the RNE, and these cells eventually adopted RNE-like morphology, we asked whether they would also adopt RNE fate. To test this, we used Vsx2, an early retinal transcription factor specific for RNE fate (*Kimura et al., 2006*). Time-lapse imaging of the transgenic line Tg(*vsx2*::GFP) expressing GFP under the *vsx2* promoter allowed us to follow RNE specification. In control embryos, GFP was primarily expressed in the invaginating RNE layer and the signal was weak in cells within the rim zone. Only when rim cells reached the inner RNE did they express GFP more prominently (*Video 20*, *Figure 6A*). However, in the ezrin knockdown condition, rim cells that accumulated outside the RNE expressed GFP ectopically at this location (*Video 21*, *Figure 6A,B*). Such ectopic expression of *vsx2* outside the optic cup

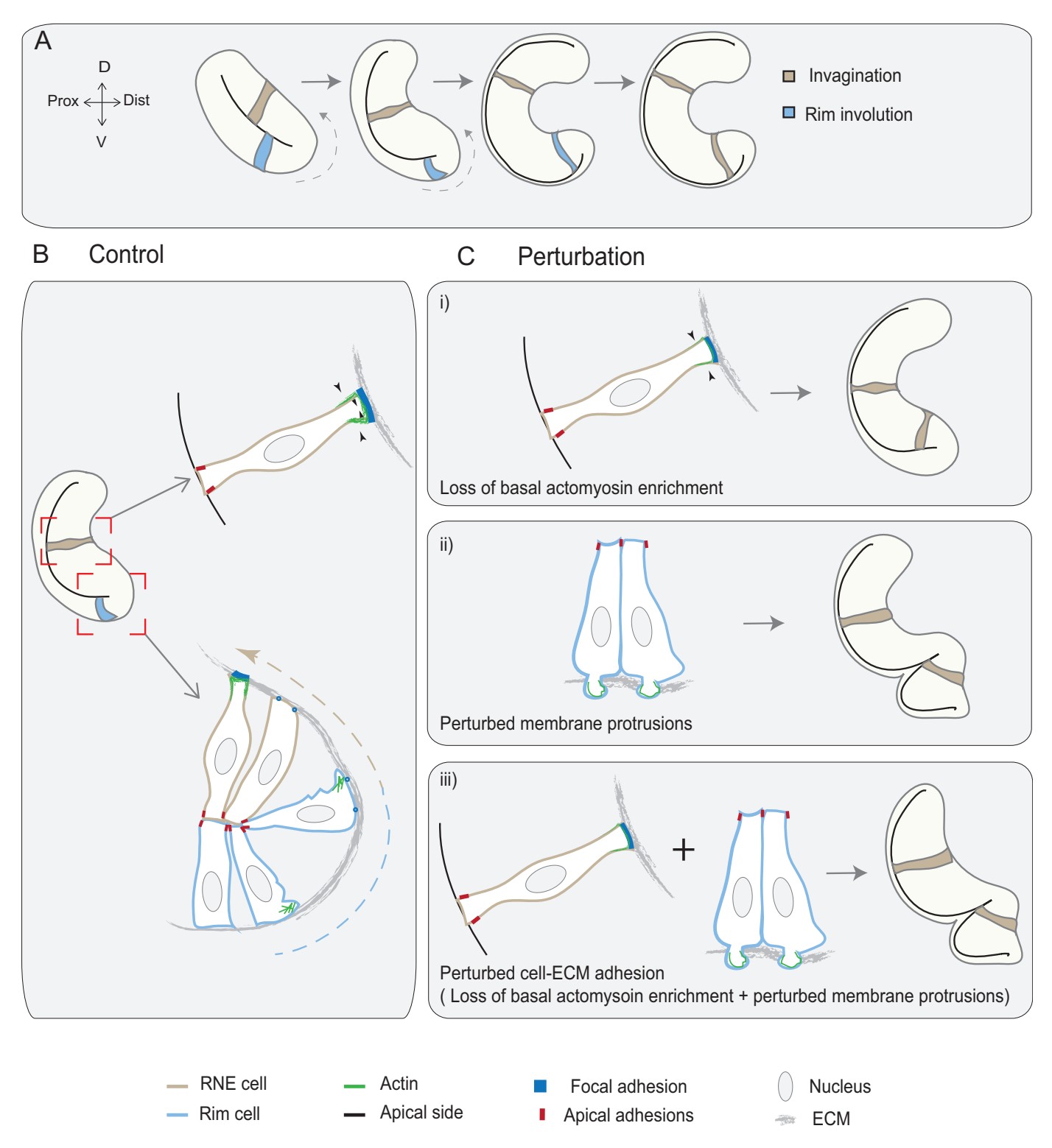

**Figure 7.** Concerted action of basal cell area shrinkage and rim involution shapes the hemispheric RNE. (**A**) Schematic representation of RNE morphogenesis. The cells in the bilayered optic vesicle shape the RNE into a hemispheric cup. RNE cells in the distal layer (brown) contribute to invagination and prospective neuroepithelial cells (blue) undergo rim involution to reach the inner layer of the cup. (**B**) In control conditions, invagination is driven by basal area reduction that is guided by basally enriched actomyosin-driven constriction and overall compaction by increasing number of cells. Rim involution is driven by collective and directed migration of the epithelium at the rim of the developing optic cup. Protrusive

*Figure 7 continued on next page*

*Figure 7 continued*

migratory dynamics of rim cells change to adherent behavior when cells reach the inner layer. (**C**) Effect of cellular perturbations on the RNE architecture. (i) Loss of basal actomyosin enrichment slows the invagination process, which can result in a wider optic cup. (ii) Perturbation of lamellipodial membrane protrusions affects the migratory behavior and the optic cup architecture, resulting in an S-shaped optic cup. (iii) Perturbation of cell-ECM adhesion results in both loss of basal actomyosin accumulation in the invaginating zone and perturbed lamellipodial membrane protrusions in the rim zone. Such combined effect leads to a severe optic cup phenotype.

was also observed in laminin and opo morphants. Here, cells failed to migrate and an S-shaped cup was formed (*Figure 6B*). Thus, *vsx2* expression and RNE fate while seemingly temporally controlled were independent of correct cell position. In some ezrin morphants, these displaced RNE cells initiated a second invagination zone (*Figure 6C*). These phenotypes persisted until 36hpf, when neurogenesis in the RNE is known to start (*Weber et al., 2014*), resulting in abnormal architecture of the optic cup (*Figure 6D*). Thus, our results suggest that rim migration functions as a mechanism to ensure that initially 'misplaced' prospective RNE cells are moved to their correct location before they adopt RNE fate, making it a crucial step for all further retinal development.

## Discussion

In this study, we show that active rim involution is an important driver of optic cup formation. It occurs by collective and directed migratory behavior of the epithelium that later integrates into the invaginating RNE layer. Rim involution together with the basal constriction of invaginating RNE cells shape the epithelial fold of the optic vesicle into a hemispherical RNE that gives rise to the future retina. While defects in RNE invagination can be rescued by continued rim involution, defects of rim migration result in impaired RNE architecture. Thus, rim involution ensures translocation and timely integration of prospective neuroepithelial cells into the optic cup, where they adopt retinal fate (summarized in *Figure 7*).

### Basal RNE cell shrinkage and active rim migration together ensure efficient hemispheric RNE formation

The teleost optic vesicle is an epithelial bilayer that needs to rearrange to form the hemispheric optic cup hosting the RNE that give rise to all retinal neurons later in development. To make this rearrangement efficient over a short developmental time span, the tissue has adopted an interplay of cell shape changes supported by basal constriction of RNE cells (*Nicolás-Pérez et al., 2016*) and migration of rim cells. However, it seems that rim involution plays a dominant role in this process as it translocates a substantial number of cells into the RNE that later contribute to invagination through basal actomyosin contractility and tissue compaction. As a result, the phenotypes observed upon perturbed rim involution are more severe than when basal constriction is impaired. Similar combinations of epithelial rearrangements are at play during other morphogenesis phenomena that occur in a rather short developmental time frame. For instance, during zebrafish posterior lateral line development, timely coordination between the epithelial rearrangements of microlumen formation and primordium migration determines the spacing of rosettes and in turn the lateral line architecture (*Durdu et al., 2014*). Thus, to fully understand the dynamics of morphogenetic processes it is important to decipher how diverse cell behaviors and their relationship orchestrate tissue formation in different contexts. This is important for determining the common and differential mechanisms underlying organ formation.

### Distinct dynamics at the basal side of the epithelium drive RNE morphogenesis

Interestingly, all of the dynamic morphogenetic changes elucidated here occur at the basal side of the epithelium. In the rim region, basal cell dynamics are crucial for cell migration, whereas in the invaginating region they assist the formation of stable basal adhesion and basal constriction. Basal domain dynamics are also involved in the formation of the zebrafish midbrain-hindbrain boundary, where the ECM is important for basal shrinkage of the boundary cells (*Gutzman et al., 2008*), and for the generation of the *Drosophila* follicular epithelium, where cell-ECM adhesion plays a crucial

role in follicle rotation and egg elongation (*Haigo and Bilder, 2011*). Furthermore, also studies on in vitro *optic* cup organoids show that laminin is a crucial ECM component for successful cup formation (*Eiraku et al., 2011*). However, most studies of established systems of epithelial morphogenesis such as gastrulation movements in *Drosophila* or vertebrate neurulation (*Guillot and Lecuit, 2013*) have focused on the dynamics of the apical domain. It will therefore be important to further explore the role of dynamics of the basal domain and its interaction with the ECM during morphogenesis in different developmental contexts.

## RNE morphogenesis occurs by spatio-temporal transition of cell behaviors from a migratory to an adherent epithelial state

During development, cells rarely exhibit archetypal epithelial or mesenchymal characteristics but usually a mixture of both. This has led to the emerging concept of a continuum between the mesenchymal and epithelial state, a topic that recently gathered increasing attention (*Campbell and Casanova, 2016*; *Bernadskaya and Christiaen, 2016*). Depending on the context and the morphogenetic processes, cells utilize different states along this continuum. Rim cells represent such an example where initially cells exhibit both epithelial (apicobasal polarity and cell-cell adhesion) and mesenchymal characteristics (directed protrusive activity and dynamic cell-matrix contacts). However, once the cells involute and integrate into the invaginating RNE they stop their protrusive activity, adhere stably to the ECM and attain the full epithelial state. Thus, rim cells traverse along the continuum and modulate their cell behavior in space and time to shape the organ precursor, the RNE. Furthermore, the spatiotemporal regulation of this transition seems important as a premature transition to stably attached morphology hinders further steps of organogenesis. Similar cellular transitions are also observed at earlier stages of eye development, during optic vesicle evagination. During this event, the eye field cells that seem of more mesenchymal nature change their morphology and acquire apicobasal polarity to form the optic vesicle (*Ivanovitch et al., 2013*; *Bazin-Lopez et al., 2015*). Thus, in teleosts, the process of RNE formation from eye field to hemispherical retina utilizes gradual transitions from the mesenchymal to epithelial state at different developmental stages to efficiently shape the RNE into a hemisphere. Further research is required to understand how cell behavior is modulated spatiotemporally. Such investigations will provide insights into the transition along the continuum of states from mesenchyme to epithelium.

## Rim cells migrate collectively

We show that rim cells undergo active cell migration using dynamic lamellipodia and cell-ECM attachments while staying integrated within the epithelial sheet. Thus, rim cell migration is a previously uncharacterized form of collective epithelial cell migration. Lately various morphogenetic phenomena have been discovered to include collective epithelial migration, including the well-studied systems of border cell migration in *Drosophila*, branching morphogenesis of trachea in *Drosophila* and mammary gland development, wound healing in mammals or lateral epidermal movement during dorsal closure (*Scarpa and Mayor, 2016*; *Friedl and Gilmour, 2009*). Studies of these different systems have led to the emerging view that in each context, the cells exhibit different extent of epithelial and migratory characters, thus broadening the definition of collective epithelial migration (*Campbell and Casanova, 2016*). Interestingly, in contrast to other collective migration phenomena, such as border-cell migration in *Drosophila*, migration of posterior lateral line primordium in zebrafish or wound healing (*Scarpa and Mayor, 2016*; *Friedl and Gilmour, 2009*), rim cell migration occurs in a part of a continuous epithelial sheet and lacks specific leader cells. Nevertheless, even without leader cells, rim migration occurs in a directed manner and lamellipodia formation is highly biased towards the direction of movement. This molecular and cellular directionality could possibly emerge from the distribution of forces present in the developing optic cup and/or the spatial distribution of surrounding ECM. Future studies will need to investigate these factors to understand how they influence directionality of rim migration.

## Defects in rim migration lead to ectopic fate specification and interfere with future retinal development

Our data show that collective rim migration is indispensable for shaping the RNE, the organ precursor that gives rise to the retina. Rim migration translocates prospective neuroepithelial cells that

initially reside in the epithelial sheet outside the presumptive RNE to their correct location in the hemispheric RNE. However, when these cells do not reach the RNE in time, they change their morphology and adhere to the underlying ECM at ectopic positions generating a negative feedback, as these cells can never be translocated further. Thus, timely controlled rim cell migration is critical to coordinate spatial positioning of cells with the timing of neuroepithelial fate determination. Failure of rim movement leads to enduring defects in optic cup architecture. This is highlighted by the observation that, in severe cases, a secondary invagination zone can be formed. Our study thereby reveals that RNE fate is independent of the position of cells, suggesting that while signaling pathways and patterning molecules are very important for RNE fate specification (*Fuhrmann, 2010*), they cannot ensure proper positioning of cells if epithelial rearrangements and morphogenic movements are impaired.

## Developmental patterning of cell behaviors is a conserved feature of vertebrate eye development

The crosstalk between morphogenesis and developmental patterning is crucial for successful organ development. Such developmental pattern is not only reflected in gene expression differences but also affects the distinct cell behaviors during early eye development (*Picker et al., 2009*; *Heermann et al., 2015*). The dorsal or distal layer of the optic vesicle starts invagination whereas the ventral region contributes to rim involution, predominantly in the ventral-temporal part of the retina (*Schmitt and Dowling, 1994*; *Heermann et al., 2015*; *Picker et al., 2009*; *Kwan et al., 2012*). Consistent with this, the phenotypes we observed in rim perturbation conditions result in accumulation of rim cells specifically at the ventral-temporal side of the optic cup. In contrast, cells are much less affected at the dorsal and nasal side. Interestingly, the architecture of the teleost optic vesicle is different compared to that of higher vertebrates. Nevertheless, ventral-temporal cell movements have not only been observed in teleosts but also during *Xenopus and* chick RNE morphogenesis (*Holt, 1980*; *Kwan et al., 2012*). Furthermore, studies in mouse have indicated that the invagination of the dorsal and ventral RNE can be uncoupled (*Molotkov et al., 2006*). Therefore, we speculate that the specific cell behavior at play might depend on the species and the shape of the optic vesicle. However, the use of differential morphogenetic strategies for dorsal and ventral retinal development is conserved in all in vivo systems studied so far. It is possible that such 'pattern' of morphogenic strategies allows the segregation of cells that experience different environments. Consequently, this could help the cells to attain the RNE fate in a stepwise manner to prepare for the later retinal developmental programs. As it has been recently suggested that such patterning could be conserved during in vitro optic cup formation in organoid cultures (*Hasegawa et al., 2016*) it will in the future be very exciting to assess how far developmental and cell-behavioral patterning in retinal organoids recapitulates the in vivo scenario at different developmental stages.

# Materials and methods

## Zebrafish strains and transgenic lines

Wild type strains (WT-AB RRID:ZIRC_ZL1, WT-TL RRID:ZIRC_ZL86) and transgenic lines Tg(actb1:GFP-utrCH), (*Behrndt et al., 2012*), Tg(actb2:mCherry-Hsa.UTRN) (*Compagnon et al., 2014*), Tg(actb1:myl12.1-EGFP) (*Maître et al., 2012*), Tg(actb1:HRAS-EGFP) vu119 (*Cooper et al., 2005*), Tg(*vsx2*::GFP) (*Kimura et al., 2006*), Tg($\beta$actin:mKate2-ras) were used. Zebrafish were maintained and bred at 26.5°C. Embryos were raised at 28°C and then transferred to 21°C at around 80% epiboly to slow down development. At 8 ss embryos were transferred back and maintained henceforth at 28°C. All animal work was performed in accordance with European Union (EU) directive 2011/63/EU as well as the German Animal Welfare Act.

## Morpholino, RNA and Plasmid injections

For morpholino-mediated knockdown of gene function, the following amounts of morpholinos were injected in the yolk at one-cell stage:

0.4 ng *laminin α−1* MO 5'TCATCCTCATCTCCATCATCGCTCA3'(*Pollard et al., 2006*),

3.8 ng *ojoplano* MO 5'ggactcacccaTCAGAAATTCAGCC3' (*Martinez-Morales et al., 2009*), 1.6 ng *ezrin* MO 5'GATGTAGATGCCGATTCCTCTCGTC3' (*Link et al., 2006*)

2 ng *p53* MO 5'GCGCCATTGCTTTGCAAGAATTG3'(*Robu et al., 2007*).

See *Supplementary file 1* for the phenotypes spread and number of embryos.

For whole tissue labeling, RNA was injected at the one-cell stage. For mosaic labeling of cells, either DNA was injected at one-cell stage or RNA was injected in a single blastomere at 16- to 32-cell stage. RNA was synthesized using the Ambion mMessage mMachine kit and injected at 50–60 pg per embryo, whereas DNA was injected at 15 pg per embryo.

## Constructs

pCS2+mKate2-ras (*Weber et al., 2014*), pCS2+GFP-ras (kind gift from A. Oates), pCS2+Par3-GFP (*Tawk et al., 2007*), pCS2+GFP-UtrophinCH (*Burkel et al., 2007*), pCS2+DD-myl12b-GFP (*Norden et al., 2009*), pCS2+ EMTB-tdTomato (kind gift from D. Gilmour), βactin::mKate2-ras (*Icha et al., 2016*), pCS2+Paxillin-mKate, pCS2+ Integrinβ1b-mKate. (for cloning strategies see below).

## Drug treatments

Dechorionated embryos were incubated in the drug solutions made in E3 medium. Rockout and Aphidicolin were dissolved in DMSO. Hydroxyurea was dissolved in water. Rockout treatment was started around 13–14 ss and was used at 100 µM with 1% DMSO (by volume) in E3. During time-lapse imaging, Rockout was used at 125 µM. For inhibition of cell proliferation, embryos were treated with a mixture of 30 mM HU and 210 µM Aphidicolin. HU+Aphi treatment was started at 10ss. Equivalent amounts of DMSO were added as solvent control.

## Transplantations

The donor embryos (GFP-ras positive) were co-injected with the p53 morpholino along with Opo or ezrin morpholino. Embryos at high to sphere stage were dechorionated and some tens of cells from the donor embryos were transferred into the animal pole of the acceptors (mCherry-UtrCH positive). Transplanted embryos were then transferred to E3 medium supplemented with 100 U penicillin and streptomycin.

## Immunostaining

Embryos were fixed with 4% PFA in PBS overnight at 4°C, followed by permeabilisation using PBT (PBS with 0.8% Triton X-100). To improve permeability, embryos were trypsinized on ice for 15 min, followed by blocking with 10% normal goat serum and incubated in the primary antibody mix with 1% NGS in PBT for 60 h at 4°C. After washing, embryos were incubated with secondary antibody mix with 1% NGS in PBT for 60 h. The embryos were mounted either in agarose or in 80% glycerol.

The following dilutions were used. Primary antibodies: 1:50 anti-phospho-myosin (RRID:AB_330248, Cell signaling 3671), 1:100 anti-laminin (RRID:AB_477163, Sigma L-9393), 1:100 anti-chondroitin sulphate CS-56 (RRID:AB_298176, Abcam ab11570), 1:100 anti-fibronectin (RRID:AB_476976, Sigma F3648), 1:500 anti-pH3 (RRID:AB_2295065, Abcam ab10543), 1:500 anti-tRFP (RRID:AB_2571743, Evrogen AB233), 1:100 anti-GFP (RRID:AB_94936, Milipore MAB3580), 1:200 anti-PKC ζ C-20 (RRID:AB_2300359, Santa Cruz sc-216).

Secondary antibodies and fluorescent markers: 1:500 Alexa Fluor 488 anti-Rabbit (RRID:AB_141708, Invitrogen A21206), 1:500 Alexa Fluor 568 anti-Rabbit (RRID:AB_2534017, Thermo Fischer Scientific A10042), 1:500 Alexa Fluor 647 anti-Rabbit (RRID:AB_141775, Invitrogen A21245), 1:500 Alexa Fluor 488 anti-mouse (RRID:AB_141606, Invitrogen A21200), Alexa Fluor 594 anti-mouse (RRID:AB_141630, Invitrogen A21201), 1:500 Alexa Fluor 647 anti-rat (RRID:AB_141778, Invitrogen A21247), 1:50 Alexa Fluor 488 Phalloidin (RRID:AB_2315147, Molecular Probes A12379), 1:50 Rhodamine-Phalloidin (RRID:AB_2572408, Molecular probes R415), DAPI.

## Image acquisition
### Brightfield imaging
Embryos were anaesthetized with 0.04% MS-222 (Sigma, St. Louis, Missouri, USA) and mounted on a drop of 3% methylcellulose in E3 medium. Images were taken on a Leica M165C scope with an MC170 HD camera.

### In Vivo Time-Lapse imaging

Embryos were mounted in 0.6% low melt agarose in E3 medium on Mattek glass bottom dishes for spinning disk confocal microscopy and in a capillary for Lightsheet microscopy. Embryos were anaesthetized using 0.04% MS-222 (Sigma).

An Andor spinning disk system with a 40x silicon oil objective (NA = 1.25) was used with a heating chamber; z stacks of 90 µm were acquired with optical section of 0.7 µm every 3–5 min for about 6–7 h. For time-lapse imaging of GFP-UtrophinCH, myl12.1-EGFP transgenic embryos and paxillin-mKate injected embryos (*Videos 1*, *2* and *12*), a Zeiss Lightsheet Z.1 microscope with a Zeiss Plan-Apochromat 20x water-dipping objective (NA = 1.0) and sample chamber heated to 28°C was used.

Maximum intensity projections of a few slices were used for visualization purpose.

### Confocal scans

Imaging was performed on Zeiss LSM 710 confocal microscopes with a 40x water-immersion objective (NA = 1.2).

### Image analysis

Image analysis was performed using Fiji platform RRID:SCR_002285 (*Schindelin et al., 2012*).

### RNE cell apical and basal area analysis

Areas of the RNE cells were measured by analyzing the optical section below the developing lens placode for basal areas and apical side of the same cells for apical areas. A rectangular area was marked where the apical and basal endfeet were observable. The number of cells in this area was calculated using multipoint tool in Fiji RRID:SCR_002285 (*Schindelin et al., 2012*). Cells, which were partially overlapping in the field, were counted only on left and top border of the rectangle.

### Actomyosin distribution analysis

The average distribution of actomyosin along the apicobasal axis of the RNE was measured using a custom-made Fiji compatible Python script (Benoit Lombardot and Robert Haase, see *Source code 1*). The region of interest (ROI) was defined as a 10 µm x 10 µm x h cuboid, with h corresponding to the height of the apicobasal axis of the RNE layer. Using this ROI, an average intensity value was calculated for each point along the apicobasal axis normalized to 100. To compare across samples, the average intensities were normalized to the highest average intensity value along the axis. This was calculated for five different regions each in multiple optic cups.

### Invagination angle analysis

The invagination angle was measured manually as the angle held at the center of the cup by the inner lips of the optic cup using the angle tool in Fiji RRID:SCR_002285 (*Schindelin et al., 2012*). The angle was measured at three different central optical sections in an optic cup and the average value was used as the angle of invagination.

### Rim cell speed analysis

The basal side of the migrating cells was tracked using MTrackJ plugin in Fiji (*Meijering et al., 2012*). The cell speed was calculated as a ratio of the track length to the track duration.

### Integrin intensity analysis

Embryos injected with a mixture of ras-GFP and Integrin-mKate2 RNA or coinjected with the RNA mixture and Opo morpholino were imaged at 24hpf. Single optical sections featuring complete apico-basal length of RNE cells were imaged and chosen for analysis. Average integrin-mKate2 intensity was measured along a 20 pixel thick line marked along the apicobasal axis of RNE. An average was calculated along the line for 0–5 µm (basal) and 25–30 µm (central) region. A ratio of basal average intensity to central average intensity was calculated for each embryo. This was calculated for five different embryos each.

Statistical analysis and graphical representation were performed using the Prism software package.

## Cloning strategies

The following constructs were generated using Gateway cloning system.

### pCS2+ paxillin mKate2

Middle entry clone for zebrafish paxillin was a kind gift from Clarissa Henry (*Goody and Henry, 2010*). It was combined with mKate2 pENTR(R2-L3) (kind gift from Andrew Oates, Crick Institute, London, UK) and pCS2 Dest(R1-R3) backbone (*Villefranc et al., 2007*).

### pCS2+ integrin beta1b-mKate2

Zebrafish Integrin $\beta$1b (NM_001034987.1) was amplified from cDNA using the following primers to generate a middle entry clone without a stop codon at the end.

   5' GGGGACAAGTTTGTACAAAAAAGCAGGCTGGatggacgtaaggctgctcc 3'
   5' GGGGACCACTTTGTACAAGAAAGCTGGGTtttgccctcatatttagggttgac 3'

It was combined with mKate2 pENTR(R2-L3) (kind gift from Andrew Oates, Crick Institute, London, UK) and pCS2 Dest(R1-R3) backbone (*Villefranc et al., 2007*).

## Zebrafish transgenesis

1 nl of the mix of Tol2 plasmid Tol2-bactin::mKate2-ras (*Icha et al., 2016*) containing 20 ng/μl and Tol2 transposase RNA 30 ng/μl in ddH$_2$O was injected into the cytoplasm of one-cell stage embryos. $F_0$ embryos with observed fluorescence signal were grown to adulthood and Tg carriers were identified by outcross with wild type fish.

## Western blot

For each condition protein sample from five embryos was used and western blot performed. Following antibodies dilutions were used: 1:500 anti-laminin (RRID:AB_477163, Sigma L-9393), 1:500 anti-phospho-ERM (RRID:AB_2262427, Cell Signaling 3141), 1:10000 Anti-α-Tubulin (RRID:AB_477582, Sigma T6074), 1:20000 peroxidase conjugated anti-Mouse (RRID:AB_2340061, Jackson Immuno 315-035-003), 1:20000 peroxidase conjugated anti-Rabbit (RRID:AB_2313567, Jackson Immuno 111-035-003).

## Acknowledgements

We thank Y Bellaiche, E Knust, J Martinez-Morales, D Mateju, M Palfy, and IK Patten for helpful comments on the manuscript. We also thank the Norden lab for useful discussions, B Lombardot and R Haase for the script allowing analysis of actomyosin distribution. Specially N Iffländer as well as C Fröb, J Icha, S Kaufmann, M Palfy, I Yanakieva, the light microscopy and fish facility are thanked for experimental help. Thanks to D Gilmour, C Henry and A Oates for sharing constructs.

JS is a member of the IMPRS-CellDevoSys PhD program and supported by a fellowship by the DIGS-BB. CN was supported by the Human Frontier Science Program (CDA-00007/2011) and the German Research Foundation (DFG). The authors declare no conflict of interest.

## Additional information

### Funding

| Funder | Grant reference number | Author |
|---|---|---|
| Human Frontier Science Program | CDA-00007/2011 | Jaydeep Sidhaye<br>Caren Norden |
| Deutsche Forschungsgemeinschaft | SFB655 A25 | Jaydeep Sidhaye<br>Caren Norden |
| DIGS-BB | PhD fellowship | Jaydeep Sidhaye |

The funders had no role in study design, data collection and interpretation, or the decision to submit the work for publication.

## Author contributions

JS, Conceptualization, Resources, Data curation, Formal analysis, Investigation, Visualization, Methodology, Writing—original draft, Writing—review and editing; CN, Conceptualization, Data curation, Supervision, Funding acquisition, Investigation, Methodology, Writing—original draft, Project administration, Writing—review and editing

## Author ORCIDs

Jaydeep Sidhaye, http://orcid.org/0000-0001-7858-8105
Caren Norden, http://orcid.org/0000-0001-8835-1451

## Ethics

Animal experimentation: All animal work was performed in accordance with European Union (EU) directive 2011/63/EU as well as the German Animal Welfare Act.

## Additional files

**Supplementary files**

• Supplementary file 1. Morpholino injection data.

• Source code 1. Source code for actomyosin distribution analysis.

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
