## [Decision Letter]

Thank you for submitting your article "Collective epithelial migration drives timely morphogenesis of the retinal neuroepithelium" for consideration by *eLife*. Your article has been reviewed by three peer reviewers, one of whom is a member of our Board of Reviewing Editors, and the evaluation has been overseen by Marianne Bronner as the Senior Editor. The reviewers have opted to remain anonymous.

The reviewers have discussed the reviews with one another and the Reviewing Editor has drafted this decision to help you prepare a revised submission.

Summary:

This manuscript by Sidhaye and colleagues aims to understand the forces driving invagination of the vertebrate optic cup, using a combination of sophisticated live imaging, pharmacological reagents, and morpholino knockdown in the zebrafish system. Recent work has described and quantified basal constriction as a driver of invagination: experiments presented here suggest that other processes may also be involved. Rim cell involution, the movement of cells from the medial to lateral layer of the optic vesicle, has been observed previously: in this manuscript, the authors carry out the first detailed characterization of the cell biological and molecular mechanisms underlying this behavior, and examine its role in driving the formation of the hemispheric retinal neuroepithelium.

Essential revisions:

1) About the tissue specificity of the process. Laminin associates with fibronectin, collagen and entactin, and could be involved in multiple other processes other than providing a scaffold for migration (for example one could think that laminin is involved in maintaining apico-basal polarity and elongation of the neuroepithelial progenitor cells, before the invagination stage). Laminin may also well be involved in cell-cell signaling, and since it is expressed all around the optic vesicle, it is possible that it affects invagination indirectly. So the fact that the optic cup in the laminin morphant doesn't invaginate properly doesn't necessarily mean it is because the lamellipodia forming cells at the rim aren't binding to ECM. The Opo-MO is also an indirect way of looking at ECM-cell interactions. It is difficult to think of an easy experiment to fix this point. Without a specific candidate ECM protein (none has been suggested by the authors) – it is unclear whether the cells bind laminin directly or another ECM protein, and experiments are difficult due to the multitude of possible receptors, some of which are likely to be redundant. Given that the group is using integrin-beta1-mKate2 for imaging, performing experiments involving DN-integrin-beta1 (placed under a heat shock promoter or a rim cell specific promoter) might help to clarify this issue. Another possibility would be to do some transplantation experiments to obtain opo and laminin morphant clones in a wt retina and observe the phenotype of the morphant rim cells (eg. lamellipodia, integrin b1 and paxillin dynamics).

2) About the link between Opo and Laminin morphant phenotype to ECM-cell interactions in the rim. Time-lapse imaging experiments in a background that labels focal adhesion markers (ie. repeat the morpholino experiments in the Integrin-β1-mKate2 and Paxillin-mKate2 background and see if the short-lived foci observed in WT rim cells disappears) would really help to make the case stronger.

3) Delineating the contribution of basal constriction vs active migration. The analysis of both the Opo morphant and the Ezrin morphant would benefit from transplantation experiments followed by time-lapse imaging to overcome limitations of spatial specificity eg. Transplant Opo/Ezrin morphant cells into labelled, but otherwise WT, embryos. If only cells at the rim are actively migrating and are pulling the epithelial sheet behind them, the migration speed of the WT cells should not be affected until the first transplanted cells reach the rim.

4) The authors state that this is a "novel mode of active collective epithelial migration". Based on the data presented, it is not quite clear to me that this is really collective migration. The videos of EGFP-DN-Rac-expressing embryos would help significantly, as would determining whether the remaining protrusions in EGFP-DN-Rac-expressing cells are positive for actin and paxillin, and therefore might mediate a different mode of motility, rather than being pulled along (see below in specific points).

5) Morpholino. The authors should use mutants when available, otherwise they should accordingly moderate their conclusions.

For example, it looks like the quantity of opo morpholino is high (3.8ng). Other tissues involved in invagination of the RNE or potentially important for optic cup formation are also altered (eg. pre-lens ectoderm). Therefore, the strong phenotype observed might not be specific. When compared with the phenotype of the opo-like mutant phenotype in Martinez-Morales et al., 2009, the retina does not look like as altered as showed here. Are the controls made for the laminin and opo MO injections complete? It seems that there is either a WB or an immunostaining, not both (as mentioned in the Results section in the reference to Figure 4—figure supplement 2). There are laminin alpha 1 mutant available (Pathania et al., 2014) and ezrin2 mutant (ZIRC). Overall these mutants would complement well the morphants and set good standards in the field.

Suggested revisions:

6) The authors state in the first sentence of the Discussion: "active rim involution is the major driver of optic cup formation." The data presented still leave some question as to whether this is the main driver: it appears that there is still a significant amount of invagination when rim involution is disrupted. This language can be softened. Yet, if both basal constriction and rim involution are inhibited simultaneously, is invagination completely blocked?

7) Throughout the manuscript, the figures could use axes for orientation of the embryos.

8) Can the authors explain why they chose the Mann-Whitney test as the statistical test of choice throughout the manuscript?

9) Figure 1: Where within the optic cup were the cells that were quantified for phalloidin and phospho-myosin? A schematic of what is defined as "central" RNE would be helpful (here, and for many other points in the manuscript). Also, some labels on the schematic inset of 1F would be helpful (the yellow arrow is a bit hard to see).

10) Figure 2: The authors use a pharmacological reagent, Rockout, to test the role of actomyosin contractility downstream of the Rock pathway. Did the authors carry out a dose-response test to determine the optimal dose? This question goes with the next point: were 30% of embryos unaffected by Rockout treatment?

11) Figure 2: The categorization of phenotypes is a bit confusing. The authors have accounted for phenotypes in 70% of treated embryos; were a full 30% of embryos unaffected after 7 h treatment? Were all of these embryos included in the quantification of angle of invagination in Figure 2?

12) Figure 2: I realize that quantification of antibody staining can be problematic, but was phospho-myosin staining generally lower in the Rockout-treated embryos than in control (as would be expected from Rockout treatment)? My understanding of the normalization (from the Methods) is that a change in total staining would not be shown in these graphs (2C, D). Can the authors add some of those images to the supplement?

13) Figure 2 and Results section: "… following inhibition of proliferation, cells in the RNE displayed a slightly larger average basal footprint than control cells." What were the statistics done to determine that average RNA cell basal footprint was slightly larger?

14) Figure 3: Again, a schematic of what the authors are considering central RNE, as opposed to peripheral RNE, would be very helpful here. For example, "Such lamellipodia were only observed in the rim cells and were not seen in cells of the central RNE (Figure 3, Video 5)." Were central RNE cells included in this video?

15) Figure 3: The video shows directional protrusive activity, including ruffles that extend up the lateral edge of the cell. In Figure 3, third panel, two arrows are included to point out protrusions. Were lateral protrusions counted separately in the graph in Figure 3?

16) Figure 3, Video 6: In the Rockout-treated embryo, the cell underwent cytokinesis – is this an indication that the Rockout treatment was not sufficient to inhibit the Rock pathway?

17) Figure 3: When migration speed was calculated for DN-Rac, how mosaic was DN-Rac expression within those embryos? Was there a difference between embryos with very mosaic expression and very widespread expression? This would be more clearly indicative of collective migration: one might predict that in very mosaic conditions, cells might be "pulled" along by unaffected neighbors, whereas in embryos with very widespread expression, fewer unaffected cells would be present to pull along impaired cells.

18) Figure 3: This appears to be the key set of experiments supporting the argument that the migration during rim involution is collective. These data are difficult to evaluate without the videos; in the still images, it is hard to see how single cells are moving, as well as how affected rim involution is (in particular, in the bottom set of panels where the EGFP-DN-Rac expression is most widespread). Can the authors include the corresponding videos?

19) Figure 3: Was angle of invagination affected in embryos expressing EGFP-DN-Rac? If EGFP-DN-Rac is impairing the collective migration underlying rim involution, this seems like an important point to determine whether rim involution is a major driver of optic cup formation.

20) Figure 3, insets: It is clear that morphology of protrusions is disrupted by expression of EGFP-DN-Rac. Yet the cells still make protrusions. Is it possible that these remaining protrusions contact and engage the extracellular matrix? Do these protrusions contain actin and paxillin? This might suggest that the cells are not necessarily non-motile with neighbors "pulling" them along, but the cells still have and utilize intrinsic motility, albeit a different mode.

21) Figure 4: What percent of morpholino-injected embryos showed epithelial accumulation? Is this consistent with what is seen in mutant embryos?

22) Figure 4: Do the authors have videos of GFP-Utrophin-CH in embryos injected with either laminin α-1 or opo morpholino? Do the cells in morpholino-injected embryos ever make lamellipodial protrusions? How effective is the block of lamellipodia formation?

23) Figure 5, and Figure 5—figure supplement 1: I am concerned about the ezrin morpholino reagent. Do the embryos show signs of cell death in the head region? Is there appreciable cell death seen when imaging? Are blebs altered if cell death is inhibited?

24) Figure 6: Brightfield images are rather difficult to see. The secondary lens-like structure in particular has interesting biological implications, so perhaps slightly more in-depth characterization (visualization by confocal microscope?) would be useful.

25) About the role of Blebbing in the process. It is not clear from the MO experiments whether blebbing itself causes faulty migration, given the broad range of actions of laminin, opo and ezrin.

---

## [Author Response]

*Essential revisions:*

*1) About the tissue specificity of the process. Laminin associates with fibronectin, collagen and entactin, and could be involved in multiple other processes other than providing a scaffold for migration (for example one could think that laminin is involved in maintaining apico-basal polarity and elongation of the neuroepithelial progenitor cells, before the invagination stage). Laminin may also well be involved in cell-cell signaling, and since it is expressed all around the optic vesicle, it is possible that it affects invagination indirectly. So the fact that the optic cup in the laminin morphant doesn't invaginate properly doesn't necessarily mean it is because the lamellipodia forming cells at the rim aren't binding to ECM.*

The reviewers rightly point out the diverse roles of laminin. To clarify our interpretations, we implemented text changes and generated additional data:

A) We now further clarified in the rewritten manuscript that we believe that the laminin knock down phenotype is a result of interference with both, the invagination of RNE cells (most likely due to the observed perturbed basal actomyosin enrichment) and rim cell defects. We believe that the combination of these phenotypes leads to the more pronounced impairment of optic cup morphogenesis observed in this condition.

B) With respect to the apicobasal polarity. We now analyzed polarity markers. However, we did not observe gross level perturbations of the apicobasal polarity of the tissue that would hamper the RNE morphogenesis. As seen previously in Bryan et al., 2016, we observed ectopic aPKC staining in a small minority of delaminated cells at 24 hpf. This data has been added to the new version of the manuscript (Figure 4—figure supplement 3)

C) We performed additional experiments to clarify the role on laminin for the distribution of other ECM components. Our new data shows that indeed laminin is involved in maintaining the overall basal lamina composition and arrangement. We observed that laminin knockdown resulted in an overall enrichment of Fibronectin in the basal lamina. In control optic cups, Fibronectin was mainly localized at the rim region and under the lens placode and stained very faintly in the invaginating RNE region. However, in laminin morphants, we observe an overall enrichment of Fibronectin in the invaginating as well as rim zone. Furthermore, upon investigating Paxillin distribution in laminin morphants we observed that the rim cells respond to this altered ECM composition by less dynamic Paxillin distribution earlier outside the RNE, which most likely adds to the rim migration phenotype. Together this data shows that laminin knock down alters ECM composition and distribution which can negatively influence the migration behaviour of rim cells. This data has been implemented in the new version of the manuscript (Figure 4—figure supplement 3, Figure 5—figure supplement 1 and Video 15).

*The Opo-MO is also an indirect way of looking at ECM-cell interactions. It is difficult to think of an easy experiment to fix this point. Without a specific candidate ECM protein (none has been suggested by the authors) – it is unclear whether the cells bind laminin directly or another ECM protein, and experiments are difficult due to the multitude of possible receptors, some of which are likely to be redundant. Given that the group is using integrin-beta1-mKate2 for imaging, performing experiments involving DN-integrin-beta1 (placed under a heat shock promoter or a rim cell specific promoter) might help to clarify this issue.*

In line with the reviewers’ suggestions, we obtained the Torso-betaPS DN-Integrin construct used before in (Martinez-Morales et al., 2009). This construct was implied to impair RNE invagination when overexpressed, however in less than 30% of embryos (Martinez-Morales et al., 2009, Figure 6 and text). We subcloned the Torso-betaPS construct and put it under a heatshock promoter with a fluorescent tag. However, using this approach, we couldn’t reproduce the previously reported invagination phenotype. Instead, we observed that paxillin in cells expressing the HS-Torso-betaPS construct localizes at apical additionally to basal positions. These findings made it in our opinion difficult to interpret the phenotypes observed and we did not add this data to the manuscript.

*Another possibility would be to do some transplantation experiments to obtain opo and laminin morphant clones in a wt retina and observe the phenotype of the morphant rim cells (eg. lamellipodia, integrin b1 and paxillin dynamics).*

We performed the transplantation experiments for the Opo morphants and indeed observed basal blebbing in the morphants clones, highlighting the autonomous nature of the phenotype. This data has been added to the new version of the manuscript (Figure 5—figure supplement 1). Similar experiments are however not possible for the laminin morphants as laminin is a secreted protein and thus cell autonomy would not be expected as all control surrounding cells would generate a normal underlying laminin matrix along which the morphant cells could migrate.

*2) About the link between Opo and Laminin morphant phenotype to ECM-cell interactions in the rim. Time-lapse imaging experiments in a background that labels focal adhesion markers (ie. repeat the morpholino experiments in the Integrin-β1-mKate2 and Paxillin-mKate2 background and see if the short-lived foci observed in WT rim cells disappears) would really help to make the case stronger.*

We thank the reviewers for this suggestion. We carried out these experiments and they indeed helped to interpret the observed phenotypes. As mentioned already for point 1, when paxillin was followed in cells in laminin morphants, these cells show less dynamic Paxillin distribution already outside the RNE, which most likely additionally hampered their migration behaviour in addition to reduced lamellipodia and increased bleb formation.

In case of Opo morphants, with reduced integrin levels (Martinez-Morales et al., 2009), we observed more transient basal paxillin localization than in control cells. However, this paxillin distribution was not accompanied by lamellipodial formation, but by basal membrane blebbing. This inability to form normal lamellipodial protrusions hampers cellular migration as shown in the ezrin condition. Together this data highlights the importance of dynamic cell ECM attachments for rim involution. This data has been added to the latest version of the manuscript (Figure 5—figure supplement 1, Video 16).

*3) Delineating the contribution of basal constriction vs active migration. The analysis of both the Opo morphant and the Ezrin morphant would benefit from transplantation experiments followed by time-lapse imaging to overcome limitations of spatial specificity eg. Transplant Opo/Ezrin morphant cells into labelled, but otherwise WT, embryos. If only cells at the rim are actively migrating and are pulling the epithelial sheet behind them, the migration speed of the WT cells should not be affected until the first transplanted cells reach the rim.*

We now clarified this further in the current version of the manuscript. As mentioned already in response to point 1, we show that both basal constriction and rim involution are important for the process of RNE invagination and both processes feedback onto each other. Thus, also the overall speed of cells is a combined effect of the cell behavior distribution in the tissue and not determined by the identity of individual clones. As the transplanted minority of morphant cells are surrounded by cells that move with the same kinetics as cells in controls and also the invaginating cells, that most likely additionally help rim involution are not affected, we did not expect major differences in speed. Indeed, due to these collective effects, the mosaic morphant cells migrate almost by similar speed as control cells (data not shown) underlining once more the collective nature of the migration. We have clarified this point in the new version of the manuscript.

*4) The authors state that this is a "novel mode of active collective epithelial migration". Based on the data presented, it is not quite clear to me that this is really collective migration. The videos of EGFP-DN-Rac-expressing embryos would help significantly, as would determining whether the remaining protrusions in EGFP-DN-Rac-expressing cells are positive for actin and paxillin, and therefore might mediate a different mode of motility, rather than being pulled along (see below in specific points).*

This is a good point brought up by the reviewer. We performed further experiments with the DN-Rac condition but due to experimental and technical difficulties it was not possible to exclude that the EGFP-DN-Rac expressing cells might mediate a different mode of motility. We have thus now removed this data from the manuscript. The collective nature of cell movement is still shown however for example in the transplanted morphant cells of the opo condition that reproduce the previous DN-Rac phenotype (see answer to Point 1 and Point 3, Video 17). We also added two additional videos that make clear that while rim cells are very dynamic at the basal side, they stay connected in the epithelial sheet apically (Figure 3, Video 7 and Video 8).

*5) Morpholino. The authors should use mutants when available, otherwise they should accordingly moderate their conclusions.*

We do agree that a combination of morphant and mutant data can strengthen conclusions especially when previously unknown proteins are explored. While it is correct, that some mutants exist, we did not have them established in our fish facility and getting and growing them would have been beyond the timescale of the revision process.

However, we here used well characterize and published morpholinos, for which knock down potential has been shown in previous studies and been compared to the mutants in the case of opo and laminin-alpha1, for which mutants show very similar phenotypes as the morphants (Martinez-Morales, et al., 2009, Pollard et al., 2006). Recently, Bryan et al., 2016 also reported the optic cup morphogenesis phenotypes in the laminin-alpha1 mutant. The ezrin morpholino was used by Link et al., 2006. Here it was shown that strong ezrin depletion can result in gastrulation defects, however, we use a morpholino concentration that allows us to interfere with later developmental stages and we show that overall development and morphology of the embryos is comparable to controls (Figure 5—figure supplement 2). Nevertheless, in addition we carried out our own knockdown controls and show that in all conditions expression levels of the genes in question are affected at the protein level (see below).

*For example, it looks like the quantity of opo morpholino is high (3.8ng). Other tissues involved in invagination of the RNE or potentially important for optic cup formation are also altered (eg. pre-lens ectoderm). Therefore, the strong phenotype observed might not be specific.*

3,8ng is not a particularly high concentration in the field. Morpholino concentrations can vary quite significantly depending on target gene, (e.g. 4ng/embryo *aPKC*-lamba morpholino recapitulates heart and soul mutant phenotype (Horne-Badovinacet al., Current biology 2001), or *vsx2*MO was used at 6–10 ng/embryo to study its role in optic cup development (Gago-Rodrigues et al., Nat comm 2015). In terms of molarity, we inject 1.5 nL of 300µM morpholino dilution which is not particularly high in the field. Furthermore, we used similarly high or higher concentration of standard control morpholinos and did not observe any phenotypes.

We also ruled out that morpholinos lead to general morphological effects, as we did not observe differences in the development of the lens placode or RPE (Figure 4). To make the variation of phenotypes observed more transparent, we now added a detailed table on the frequency of the phenotypes to the current manuscript. (Material methods—source data1).

*When compared with the phenotype of the opo-like mutant phenotype in Martinez-Morales et al. 2009, the retina does not look like as altered as showed here.*

The phenotype seen in our study does phenocopy the phenotype shown for zebrafish morphants as shown in the Supplementary Figure 4 in the Martinez-Morales et al. 2009 Development paper.

Subtle differences, that we are nevertheless not aware of, might result from the differences in the medaka versus the zebrafish system. In personal communication with Juan Martinez Morales he confirmed that also his lab observed the rim phenotype, despite the fact that they did not further study it. Thus, we do believe that what we observe here is a genuine opo phenotype in zebrafish.

*Are the controls made for the laminin and opo MO injections complete? It seems that there is either a WB or an immunostaining, not both (as mentioned in the Results section in the reference to Figure 4—figure supplement 2).*

In opo morphants no antibody staining could be performed due to lack of commercial antibody. However, we now added additional controls to test the localization of integrins in this condition (Figure 4—figure supplement 2).

*There are laminin alpha 1 mutant available (Pathania et al. 2014) and ezrin2 mutant (ZIRC). Overall these mutants would complement well the morphants and set good standards in the field.*

See our response to the first point in response 5.

*Suggested revisions:*

*6) The authors state in the first sentence of the Discussion: "active rim involution is the major driver of optic cup formation." The data presented still leave some question as to whether this is the main driver: it appears that there is still a significant amount of invagination when rim involution is disrupted. This language can be softened. Yet, if both basal constriction and rim involution are inhibited simultaneously, is invagination completely blocked?*

This is an important point brought up by the reviewers and we have now rephrased the whole manuscript according to this suggestion. Along these lines we also now cite the recently published study from the Martinez-Moralez lab, Nicolas-Perez et al., 2016. We further emphasize that both, invagination and rim involution play important roles in RNE formation. However, our data clearly shows that rim involution is a crucial player, as it contributes cells to the inner RNE that in turn directly contribute to invagination via the established actin and myosin basal bias.

We further show that the laminin knock down leads to exactly the scenario the reviewers speculate on. Upon laminin knock down both, invagination and rim migration are blocked, which is also the reason why this condition leads to the most severe phenotypes. See also our answers to Point 1, 2 and 3.

*7) Throughout the manuscript, the figures could use axes for orientation of the embryos.*

Following the reviewers’ suggestion, we have incorporated these axes in all our figures.

*8) Can the authors explain why they chose the Mann-Whitney test as the statistical test of choice throughout the manuscript?*

We chose Mann-Whitney as the statistical test as it is a non-parametric test. Unlike the *t*-test it does not need the assumption of normal distributions and thus can be used on broad range of data samples.

*9) Figure 1: Where within the optic cup were the cells that were quantified for phalloidin and phospho-myosin? A schematic of what is defined as "central" RNE would be helpful (here, and for many other points in the manuscript). Also, some labels on the schematic inset of 1F would be helpful (the yellow arrow is a bit hard to see).*

We apologize for the confusion the term ‘central’ retina caused. We intended to refer to the invaginating cells. We have changed this throughout the manuscript. Thus, the cells quantified for actomyosin distribution were the cells in the invagination zone. We have added a schematic to the supplementary figure (Figure 1—figure supplement 1).

*10) Figure 2: The authors use a pharmacological reagent, Rockout, to test the role of actomyosin contractility downstream of the Rock pathway. Did the authors carry out a dose-response test to determine the optimal dose? This question goes with the next point: were 30% of embryos unaffected by Rockout treatment?*

Yes, we indeed carried out a dose-response curve. At 50µM we did not observe any obvious morphological phenotype. A 150µM treatment however resulted in severe cell death in the embryo and at 200µm development was blocked completely and the embryos died. Therefore, we chose a 100µM concentration. The 30% embryos not reflected in the figures showed severe phenotypes with developmental delay. This data was added to the table [Supplementary-material SD4-data].

*11) Figure 2: The categorization of phenotypes is a bit confusing. The authors have accounted for phenotypes in 70% of treated embryos; were a full 30% of embryos unaffected after 7 h treatment? Were all of these embryos included in the quantification of angle of invagination in Figure 2?*

As mentioned above on average 30% embryos showed severe morphological defects not related to OCM and they were not considered for angle analysis.

*12) Figure 2: I realize that quantification of antibody staining can be problematic, but was phospho-myosin staining generally lower in the Rockout-treated embryos than in control (as would be expected from Rockout treatment)? My understanding of the normalization (from the Methods) is that a change in total staining would not be shown in these graphs (2C, D). Can the authors add some of those images to the supplement?*

We thank the reviewers for the suggestion. We have added the image panel in the supplement (Figure 2—figure supplement 1)

*13) Figure 2 and Results section: "… following inhibition of proliferation, cells in the RNE displayed a slightly larger average basal footprint than control cells." What were the statistics done to determine that average RNA cell basal footprint was slightly larger?*

We thank the reviewers for pointing it out, indeed the difference is statistically significant. We have now added the results of the statistical test in the figures directly.

*14) Figure 3: Again, a schematic of what the authors are considering central RNE, as opposed to peripheral RNE, would be very helpful here. For example, "Such lamellipodia were only observed in the rim cells and were not seen in cells of the central RNE (Figure 3, Video 5)." Were central RNE cells included in this video?*

As mentioned in answer to point 9, we apologize for the confusion with the term ‘central’ retina, we intended to refer to the invaginating cells. We have changed this throughout the manuscript. The rim cells after undergoing involution contributed to invagination. Thus, they are also included in the figure/video and marked by yellow arrowhead.

*15) Figure 3: The video shows directional protrusive activity, including ruffles that extend up the lateral edge of the cell. In Figure 3, third panel, two arrows are included to point out protrusions. Were lateral protrusions counted separately in the graph in Figure 3?*

Thanks for pointing it out. Our initial analysis included all protrusions including the ruffles/lateral protrusions. According to the reviewers’ suggestion, we reanalyzed the data and counted only the actin rich membrane protrusions at the base of cells. However, this analysis shows the same results, with protrusive preference at the leading edge. We replaced the previous analysis with the new analysis in (Figure 3).

*16) Figure 3, Video 6: In the Rockout-treated embryo, the cell underwent cytokinesis – is this an indication that the Rockout treatment was not sufficient to inhibit the Rock pathway?*

We would like to clarify that the purpose of the experiment was to understand the role of basal actomyosin enrichment in RNE morphogenesis. The 100µM dose allowed us to disrupt the basal actomyosin enrichment without severely affecting tissue homeostasis, proliferation or overall development. See also our answer to Points 10 and 11.

*17) Figure 3: When migration speed was calculated for DN-Rac, how mosaic was DN-Rac expression within those embryos? Was there a difference between embryos with very mosaic expression and very widespread expression? This would be more clearly indicative of collective migration: one might predict that in very mosaic conditions, cells might be "pulled" along by unaffected neighbors, whereas in embryos with very widespread expression, fewer unaffected cells would be present to pull along impaired cells.*

As mentioned above, we have removed the DN-Rac data in the latest version of the manuscript.

*18) Figure 3: This appears to be the key set of experiments supporting the argument that the migration during rim involution is collective. These data are difficult to evaluate without the videos; in the still images, it is hard to see how single cells are moving, as well as how affected rim involution is (in particular, in the bottom set of panels where the EGFP-DN-Rac expression is most widespread). Can the authors include the corresponding videos?*

As mentioned above, we have removed the DN-Rac data in the latest version of the manuscript.

*19) Figure 3: Was angle of invagination affected in embryos expressing EGFP-DN-Rac? If EGFP-DN-Rac is impairing the collective migration underlying rim involution, this seems like an important point to determine whether rim involution is a major driver of optic cup formation.*

As mentioned above, we have removed the DN-Rac data in the latest version of the manuscript.

*20) Figure 3, insets: It is clear that morphology of protrusions is disrupted by expression of EGFP-DN-Rac. Yet the cells still make protrusions. Is it possible that these remaining protrusions contact and engage the extracellular matrix? Do these protrusions contain actin and paxillin? This might suggest that the cells are not necessarily non-motile with neighbors "pulling" them along, but the cells still have and utilize intrinsic motility, albeit a different mode.*

As mentioned above, we thank the reviewers for raising this possibility. For reasons explained in answer to Point 4 we have removed the DN-Rac data in the latest version of the manuscript.

*21) Figure 4: What percent of morpholino-injected embryos showed epithelial accumulation? Is this consistent with what is seen in mutant embryos?*

We have now added a table with the numbers. See Material methods—source data 1.

*22) Figure 4: Do the authors have videos of GFP-Utrophin-CH in embryos injected with either laminin α-1 or opo morpholino? Do the cells in morpholino-injected embryos ever make lamellipodial protrusions? How effective is the block of lamellipodia formation?*

We have added the data (Figure 5—figure supplement 1, Video 15, Video 16) for paxillin and membrane labeled cells in both the conditions. We see a drastic effect on the lamellipodia formation in laminin and Opo conditions. This effect is more pronounced at the outermost point at the rim of the cup. See also our answer to Point 2.

*23) Figure 5, and Figure 5—figure supplement 1: I am concerned about the ezrin morpholino reagent. Do the embryos show signs of cell death in the head region? Is there appreciable cell death seen when imaging? Are blebs altered if cell death is inhibited?*

In response to concern raised by the reviewers, we performed caspase 3 staining on the ezrin morphants. However, we did not observe significant rates of cell death in the developing RNE.

Author response image 1.**DOI:**
http://dx.doi.org/10.7554/eLife.22689.056

Furthermore, in the transplantation experiments, cells were co-injected with the ezrin and p53 morpholino as it is known that otherwise hosts embryos often select against transplanted cells. However, ezrin morphant cells nevertheless formed blebs arguing against a direct role for cell death in bleb formation.

*24) Figure 6: Brightfield images are rather difficult to see. The secondary lens-like structure in particular has interesting biological implications, so perhaps slightly more in-depth characterization (visualization by confocal microscope?) would be useful.*

As for us this was more a peculiar observation that occurred too rarely to follow up upon, we did not want to emphasize this point and now removed this sentence from the manuscript.

*23) About the role of Blebbing in the process. It is not clear from the MO experiments whether blebbing itself causes faulty migration, given the broad range of actions of laminin, opo and ezrin.*

We apologize in case this was not clear enough in the previous version of the manuscript. We have clarified this in the current version of the manuscript. We propose that it is the perturbation of the usual protrusive activity and cell-ECM attachment that results in failed rim migration, blebbing is a cellular response observed under this condition.